# Scaling effects of fixed-wing ground-generation airborne wind energy systems

Markus Sommerfeld[1], Martin Dörenkämper[2], Jochem De Schutter[3], and Curran Crawford[1]

[1]Institute for Integrated Energy Systems, University of Victoria, British Columbia, Canada
[2]Fraunhofer Institute for Wind Energy Systems (IWES), Oldenburg, Germany
[3]Systems Control and Optimization Laboratory IMTEK, University of Freiburg, Germany

**Correspondence:** Markus Sommerfeld (msommerf@uvic.ca)

**Abstract.** While some airborne wind energy system (AWES) companies aim at small, temporary or remote off-grid markets, others aim at utility-scale, multi-megawatt integration into the electricity grid. This study investigates the scaling effects of single-wing, ground-generation AWES from small to utility-scale systems, subject to realistic 10-minute, onshore and offshore wind conditions derived from a numerical mesoscale weather research and forecasting (WRF) model. To reduce computational cost, vertical wind velocity profiles are grouped into 10 clusters using k-means clustering. Three representative profiles from each cluster are implemented into a nonlinear AWES optimal control model to determine power-optimal trajectories. We compare the effects of three different aircraft masses and two sets of nonlinear aerodynamic coefficients for aircraft with wing areas ranging from 10 to $150 \ \mathrm{m}^2$ on operating parameters and flight trajectories. We predict size and mass dependent AWES power curves, annual energy production (AEP) and capacity factor (cf) and compare them to a quasi-steady state reference model. Instantaneous force, tether reeling speed and power fluctuations as well as power losses associated with tether drag and system mass are quantified.

# 1 Introduction

Airborne wind energy systems (AWESs) harvest wind energy from stronger and less turbulent winds at mid-altitude, here defined as heights above 100 m and below 1500 m. These beneficial conditions promise more reliable and stable wind power generation compared to conventional wind turbines (WTs) at lower altitudes. The light, tower-less design allows for mobile deployment and reduces the capital cost of AWESs (Lunney et al., 2017). These kite-inspired systems consist of one or more autonomous aircraft which are connected to a ground station via one or more tethers. While various designs are investigated, two major crosswind concepts are currently considered by industry: the ground-generation also referred to as pumping-mode, and on-board-generation also referred to as drag-mode. On-board-generation AWES carry additional weight with the on-board generator and propeller mass, as well as the heavier, conductive tether. This study focuses on the cyclic two-phase, ground-generation concept, as it is currently the main concept pursued by industry.

Ground-generation AWES generate power during the reel-out phases while the wing generates large lift forces and pulls the tether from a drum. Various companies propose different reel-out pattern trajectories such as figure of eight or circular spirals, which is investigated in this research. During the following reel-in phases a fraction of the previously generated energy is consumed to return the aircraft back to its initial position and restart the cycle (Luchsinger, 2013). The upward and downward motion during the production phases are called pumping cycles. The power generated by such systems is inherently oscillating which could be offset using multiple devices in a wind farm setup or buffering the energy before feeding it into the grid (Malz et al., 2018; Faggiani and Schmehl, 2018).

Over the last years, two main AWES applications emerged. The first makes use of the mobile nature of the technology which allows the deployment in inaccessible or remote places such as temporary mines or remote off-grid communities as these locations often rely on expensive diesel generators (SkySails Group GmbH; Kitepower B.V.). The second is grid-scale integration of AWES, which requires upscaling the systems to compete with fossil and established renewable energy sources in the energy market. One example is (Ampyx) which aims to re-power decommissioned offshore wind farms or deploy floating platforms (offshorewind.biz), expecting higher energy yield due to better wind conditions, which in combination with advantageous design choices lead to lower levelized cost of electricity. Additionally, setting up AWES offshore allows for safer operation and is likely to be socially more accepted (Ellis and Ferraro, 2016).

Determining realistic performance of AWES is challenging as the flight trajectory depends on many variables that are not represented in simple models. Wind velocity profiles, aerodynamic coefficients, tether drag, aircraft mass and size affect the flight trajectory and therefore also the generated power. Using an optimization algorithm, it is possible to account for these variables and determine optimal AWES performance.

We therefore investigate the scalability and design space of small to large-scale AWES, both offshore and onshore. Depending on the aircraft wing surface area, aerodynamic coefficients and the tether diameter, rated power ranges from $\overline{P}_{\text{rated}} = 145$ to 199 kW for $A = 10 \text{ m}^2$ and $\overline{P}_{\text{rated}} = 2000$ kW to 3400 kW for $A = 150 \text{ m}^2$. Rated power is defined as the maximum cycle average power that can be achieved with a specific design. We compare the optimal system performance of different aircraft masses for representative onshore and offshore wind conditions.

The power output of an AWES not only depends on the wing size, but also the prevalent wind velocity profile shape and magnitude which result in distinct trajectories and operating altitudes. Therefore, a representative wind data set up to mid-altitudes,here defined as heights above 100 m and below 1500 m, is necessary to determine realistic AWES performance. This study relies on mesoscale numerical weather prediction models such as the Weather Research and Forecasting (WRF) model,

which is well known for conventional WT siting applications (Salvação and Guedes Soares, 2018; Dörenkämper et al., 2020), as measuring wind conditions at mid-altitudes is difficult due to reduced data availability aloft (Sommerfeld et al., 2019a). In comparison to the commonly used logarithmic wind speed profile, the WRF-derived set of wind data includes the wind direction rotation with height and the complex range of profile shapes emerging from atmospheric stability. This includes almost constant wind velocity profiles associated with unsteady stratification, high sheer wind velocity profiles resulting from stable conditions,

as well as non-monotonic wind velocity profiles including low level jets (LLJs). To reduce the computational cost, 10-minute average wind speed profiles are clustered using the k-means clustering method described in Sommerfeld et al. (2020). We compare AWES performance for an onshore location in northern Germany near Pritzwalk (Sommerfeld et al., 2019b) and an offshore location at the FINO3 research platform in the North Sea. These clustered wind conditions are implemented into the awebox (De Schutter et al., 2020) optimization framework which computes periodic flight trajectories that maximize average

mechanical power output.

In comparison to our previous studies (Sommerfeld et al., 2020), which derived onshore and offshore AWES power curves, this paper explores the AWES design space from small to utility-scale. We aim at setting up-scaling design and mass targets, instead of a detailed system design. While other studies rely on simplified logarithmic wind speed profiles (De Schutter et al., 2019), high resolution large eddy simulation (LES) (Haas et al., 2019) or reanalysis data sets (Schelbergen et al., 2020) to

investigate general behavior, performance, trajectory or wake effects, we optimize AWES trajectory subject to realistic 10 minute mesoscale wind data, which allows better optimal performance prediction.

The main contribution is the presentation of aerodynamic, mass and size scaling effects on representative ground-generation AWES subject to realistic wind conditions and operating constraints. The described results allow an informed decision-making regarding location-specific design, power estimation and scaling limitations.

The structure of the paper is as follows. Section 2 summarizes the onshore and offshore wind resource as well as the clustering results. Section 3 briefly introduces the AWES model and optimization method as well as the implemented constraints and initialization. Section 4 compares the results for six AWES sizes with three different mass scaling assumptions and two sets of nonlinear aerodynamic coefficients. We present, among other things, flight trajectories, power curves and annual energy production estimates for a representative onshore and offshore location. Finally, Section 5 concludes the article with an outlook

and motivation for future work to continue to advance AWES towards commercial reality.

## 2 Wind conditions

This study considers representative 10 min onshore (northern Germany, lat: $53°10'47.00''$N, lon: $12°11'20.98''$E) and offshore (FINO3 research platform, lat: $55°11,7'$N, lon: $7°9,5'$ E) wind data derived from 12 months of WRF simulations each. Both locations are highlighted by a black dot in Figure 1.

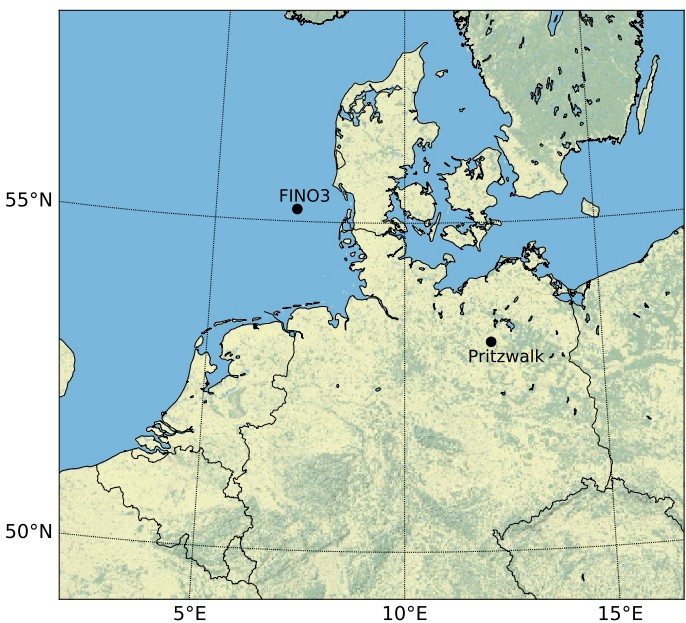

**Figure 1.** Topographic map of northern Germany with the representative onshore (Pritzwalk) and offshore (FINO3) locations highlighted with black dots and labeled.

Both horizontal velocity components of the resulting mesoscale wind data set are partitioned using a *k*-means clustering algorithm (Pedregosa et al., 2011). According to previous investigations (Sommerfeld et al., 2020), a small number of clusters with few representative profiles per cluster yield good power and AEP estimates at reasonable computational cost. Therefore, the wind velocity profiles are grouped into $k = 10$ clusters from which the 5th, 50th and 95th percentile (sorted by the average wind speed between 100 and 400 m) are implemented into the optimization algorithm as design points to cover the entire annual wind regime.

The resulting average wind velocity profiles for each of the ten clusters, also known as centroids, are shown in Figure 2 (a) and (b). For presentation purposes, only each centroid's wind speed magnitude, colored according to average wind speed up to 500 m, is shown. The complete set of clustered profiles are shown in grey. The cluster average wind profile shapes

show wind shears typically associated with unstable and stable atmospheric conditions. They follow expected location-specific

trends with lower wind shear and higher wind speeds offshore (right) in comparison to onshore (left). The associated, color-coded annual centroid frequency is shown in Figure 2 (c), (d). The diagrams in Figure 2 (e) and (f) illustrate the wind speed probability distribution at a reference height of $100 \leq z \leq 400$ m. We chose this reference height as a proxy for the wind speed at operating altitude, because an a priori estimation is impossible, and onshore and offshore power curves are almost identical using this reference wind speed. For a detailed description of the WRF model and setup, the clustering process as well as the

correlation between clusters and stability conditions see Sommerfeld et al. (2019b, 2020). Recent consensus among the AWES community defined the reference height as the pattern trajectory height, which is the expected or actual time-averaged height during the reel-out (power production) phase (Airborne Wind Europe).

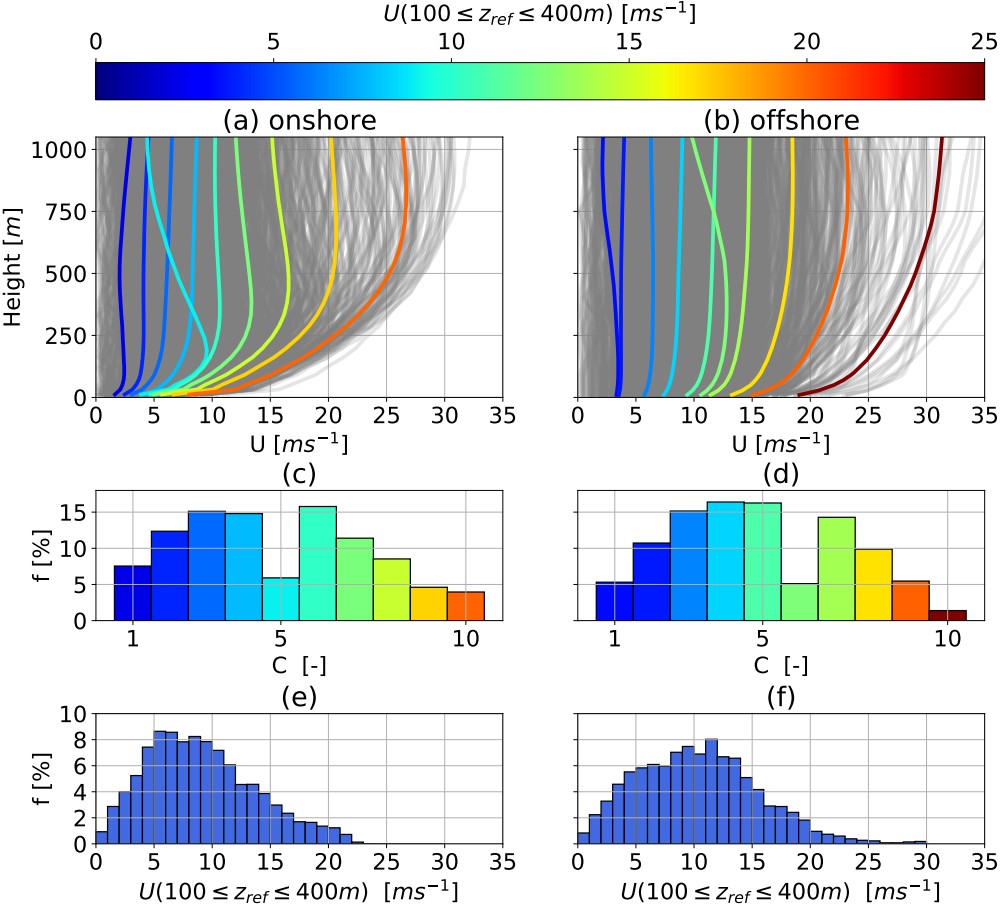

**Figure 2.** k-means clustered ($k = 10$) onshore (left column) and offshore (right column) annual cluster average wind speed profile centroids (a) and (b). The range of WRF-simulated wind speed profiles is depicted in grey. The centroids are sorted, labeled and colored coded according to average wind speed up to 500 m. The corresponding cluster frequency $f$ for each cluster $C$ is shown in (c),(d). The histograms in (e) and (f) show the wind speed probability distribution at a reference height of $100 \leq z \leq 400$ m.

## 3    Trajectory optimization model

Investigating the AWES scaling potential not only requires understanding of wind conditions at higher altitudes, but also of power production, which is intrinsically linked to the aircraft's flight dynamics, as the AWES never reaches a steady state over the course of a power cycle. Forces and moments continuously change due to transition between reeling in and out within each pumping cycle as well as the changes in flight direction inherent to typical flight patterns, such as figures of eight or circular spirals, during the production phase. Additionally, constantly changing wind conditions over a vast height range require the aircraft to adapt its trajectory. Hence, power output estimation based on steady-state simplifications only give a rough estimate, but cannot describe the variation of system parameters or operating trajectory which determine power production, particularly for realistic, non-monotonic wind profiles. Therefore, we make use of optimal control methods to compute power-optimal flight trajectories, that satisfy realistic operational constraints such as flight envelope and structural system limits. The dynamic equations therefore ensure physically realisable operating conditions, and those optimizations that fail to find a feasible solution identify cases where flight is infeasible, e.g. an airframe that is too heavy in a low-wind condition. We compare the optimization results to a simplified quasi steady-state engineering AWES model (QSM) similar to van der Vlugt et al. (2019) and Schmehl et al. (2013) (Subsection 3.2) to verify our results and to highlight the difference between both models.

### 3.1    Model overview

We compute ground-generation AWES pumping cycles by solving a periodic optimal control problem which maximizes the cycle-average power output $\overline{P}$. In periodic optimal control, the system state at the initial and final time of the trajectory must be equal, but are chosen freely by the optimizer. This methodology, implemented in the open-source software framework `awebox` (De Schutter et al., 2020), is used to generate power-optimal trajectories for single-wing ground-generation AWES sizes with variable wing area, mass and aerodynamic performance.

The AWES model considers a 6 degree of freedom (DOF) fixed-wing aircraft model with pre-computed quadratic lift, to account for stall effects, drag and pitch moment coefficients, which is controlled via aileron, elevator and rudder deflection rates. For this scaling study, the Ampyx Power AP2 reference model (Ampyx; Licitra, 2018; Malz et al., 2019) serves as a base from which the aircraft size and mass as well as aerodynamic coefficients are scaled (Sections 3.4 and 3.6). It should be noted that we used the AP2 parameters as they were available, but the mass and aerodynamic coefficients of future large-scale AWES might be quite different, because the AP2 was designed as a concept demonstrator and does not represent an optimized commercial large-scale system. In any case we later carry out sensitivity studies on the main system parameters to encompass the range of key performance parameters that might be achieved by detailed engineering of up-scaled concepts.

While the ground station dynamics are not explicitly modeled, constraints on tether reeling speed, acceleration and jerk are implemented to ensure a realistic operating envelope. The tether diameter $d$ has been chosen such that maximum average cycle power is achieved at an approximate wind speed of $10 \, \mathrm{ms}^{-1}$.

For a more detailed description of the model and the optimization algorithm see Sommerfeld et al. (2020); Leuthold et al. (2018); De Schutter et al. (2019); Bronnenmeyer (2018); Horn et al. (2013); Haas et al. (2019).

## 3.2 Quasi-steady state reference model

To contextualize the optimization results, a quasi-steady state model (QSM) based on ideal crosswind operation (Loyd, 1980) is introduced. This model has been generalized by Schmehl et al. (2013) to include losses arising from misalignment of the tether and wind velocity vector. The aircraft position is described in the spherical coordinates by the distance from the ground station, the elevation angle $\varepsilon$ and azimuth angle $\phi$ relative to the wind velocity vector. It neglects aircraft and tether mass and assumes a quasi-steady flight state, with the wing moving cross-wind with zero azimuth angle $\phi = 0$. Dividing the tether reeling speed $\dot{l}$ by the wind speed defines the reeling factor

$$f = \dot{l}/U. \tag{1}$$

with an optimal value of $f_\mathrm{opt} = 1/3 \cos\varepsilon \cos\phi$ (Argatov et al., 2009). Equation (2) estimates maximum power $P_\mathrm{max}$ as a function of wind speed $U$ at altitude $z$ and the resultant aerodynamic force coefficient $c_\mathrm{R}$ (Equation (3)), which is calculated from the aerodynamic lift $c_\mathrm{L}$ and total drag coefficient $c_\mathrm{D,total}$ of all airborne components.

$$P_\mathrm{max} = \frac{\rho_\mathrm{air}(z)}{2} U(z)^3 c_\mathrm{R} \left( \frac{c_\mathrm{R}}{c_\mathrm{D,total}} \right)^2 f_\mathrm{opt} \left( \cos\varepsilon \cos\phi - f_\mathrm{opt} \right)^2 \tag{2}$$

$$c_\mathrm{R} = \sqrt{c_\mathrm{L}^2 + c_\mathrm{D,total}^2} \tag{3}$$

Tether drag is included in $c_\mathrm{D,total}$ according to Equation (5) (Hoerner, 1965).

Increasing the power output $P_\mathrm{max}$ can be achieved by improving $c_\mathrm{R}^3/c_\mathrm{D,total}^2$ and wind speed $U$ at height $z$ as well as tether length $l$, which determine the elevation angle $\varepsilon = \arcsin(\frac{z}{l})$ and tether associated losses. A linear approximation of the standard atmosphere yields air density $\rho_\mathrm{air}(z)$ at altitude $z$ (Champion et al., 1985)

$$\rho_\mathrm{air}(z) = 1.225 \ \mathrm{kgm}^{-3} - 0.00011 \ \mathrm{kgm}^{-4} z. \tag{4}$$

The total drag coefficient $c_\mathrm{D,total}$ represents the aerodynamic drag of the entire AWES. It is derived from the tether diameter $d$, tether length $l$, the wing area $A$, as well as the aerodynamic drag coefficient of the tether $c_\mathrm{D,tether}$ and wing $c_\mathrm{D,wing}$, which depends on the angle of attack and the shape of the wing. We consider a cylindrical tether with constant diameter and an aerodynamic tether drag coefficient $c_\mathrm{D,tether} = 1.0$. The coefficient could even be higher for braided tethers. For the sake of simplicity, tether inclination with respect to the wind direction is not considered in the drag calculation, which leads to an over estimation. A more accurate tether model would further include the wind speed variation with height. Assuming a uniform

wind field, the line integral along the tether results in a total effective drag coefficient of (Houska and Diehl, 2007; Argatov and Silvennoinen, 2013; van der Vlugt et al., 2019; Schmehl et al., 2013):

$$c_{\mathrm{D,total}} = c_{\mathrm{D,wing}} + \frac{1}{4} \frac{d\,l}{A} c_{\mathrm{D,tether}} \tag{5}$$

Both the QSM and the optimization model are subject to the same constraints. the optimal power of the QSM is estimated by varying tether length up to 2000 m for every given wind profile (Section 3.3) and applying the above described tether drag and elevation losses. The same minimal operating altitude as for the optimization model is enforced. The QSM-predicted power used for reference in Subsection 4.3 is the highest power for a given wind profile. Therefore, optimal operating height is the height at which the highest power is calculated, see previous publication (Sommerfeld et al., 2020).

### 3.3 Wind boundary condition

The 2D horizontal wind velocity profiles are partitioned into $k = 10$ clusters. Three representative profiles from each cluster as well as each cluster's centroid, rotated such that the main wind direction points in positive $x$-direction and the transverse velocity component points in positive $y$-direction, are implemented into the optimization algorithm as boundary conditions. This assumes that the investigated AWES can operate independent of wind direction and are not restricted to a certain direction. This way the main wind direction of every profile points in the same direction, simplifying the comparison between different wind velocity profiles. We interpolate the $x$-direction component $u$ and $y$-direction component $v$ using Lagrange polynomials to obtain a twice continuously differentiable function representation of the wind velocity profiles, which is necessary to formulate an optimal control problem that can be solved with the gradient-based nonlinear programming (NLP) solver IPOPT (Wächter and Laird).

### 3.4 Aircraft scaling

Aircraft mass $m$ and inertia $\mathbf{J}$ are scaled relative to the Ampyx AP2 reference model (Licitra, 2018; Malz et al., 2019; Ampyx) according to simplified geometric scaling laws relative to wing span $b_{\mathrm{scaled}}$ (Equations (6) and (7)).

$$m_{\mathrm{scaled}} = m_{\mathrm{ref}} \left( \frac{b_{\mathrm{scaled}}}{b_{\mathrm{ref}}} \right)^{\kappa} \tag{6}$$

$$\mathbf{J}_{\mathrm{scaled}} = \mathbf{J}_{\mathrm{ref}} \left( \frac{b_{\mathrm{scaled}}}{b_{\mathrm{ref}}} \right)^{\kappa+2} \tag{7}$$

We investigate the impact of positive and negative scaling effects by varying the mass scaling exponents $\kappa$ between 2.7 and 3.3. An exponent of 3 represents pure geometric scaling (North et al., 2007) according to the square-cube law, while $\kappa = 2.7$ implies positive scaling effects and weight savings with size, while $\kappa = 3.3$ assumes negative scaling. A review of available literature shows that anticipated AWES scaling exponents vary between $\kappa = 2.2 - 2.6$ (grey area), shown in Figure 3.

Makani's published technical reports describe their "M600 SN6" as well as their MX2 (*Oktoberkite*) design, which is a redesign of the M600 airframe to overcome some of its shortcomings and produce $\overline{P}_{\mathrm{MX2}} = 600$ kW at a wind speed of $U_{\mathrm{MX2-ref}} = 11$ ms$^{-1}$ at operating height (Echeverri et al., 2020). Note that Makani's on-board-generation concept is inher-
ently heavier than the ground-generation concept considered here, because of propellers, generators and supporting structures onboard the aircraft. The original M600 was designed for a mass of 919 kg. The built M600 had a wing area of $A = 32.9$ m$^2$ and a mass of $m_{\mathrm{M600}} = 1730.8$ kg which is almost double the design value. If we scale the AP2 reference aircraft to the same wing area and mass, the corresponding mass scaling exponent is $\kappa = 3.23$. The airframe of the improved MX2 design aimed at $m_{\mathrm{MX2}} = 1852$ kg for a wing area of $A_{\mathrm{MX2}} = 54$ m$^2$, equivalent to $\kappa = 2.719$ relative to the AP2 reference. Similarly, wind
turbine (WT) mass scales with an exponent slightly below 3 based on rotor diameter (Fingersh et al., 2006).

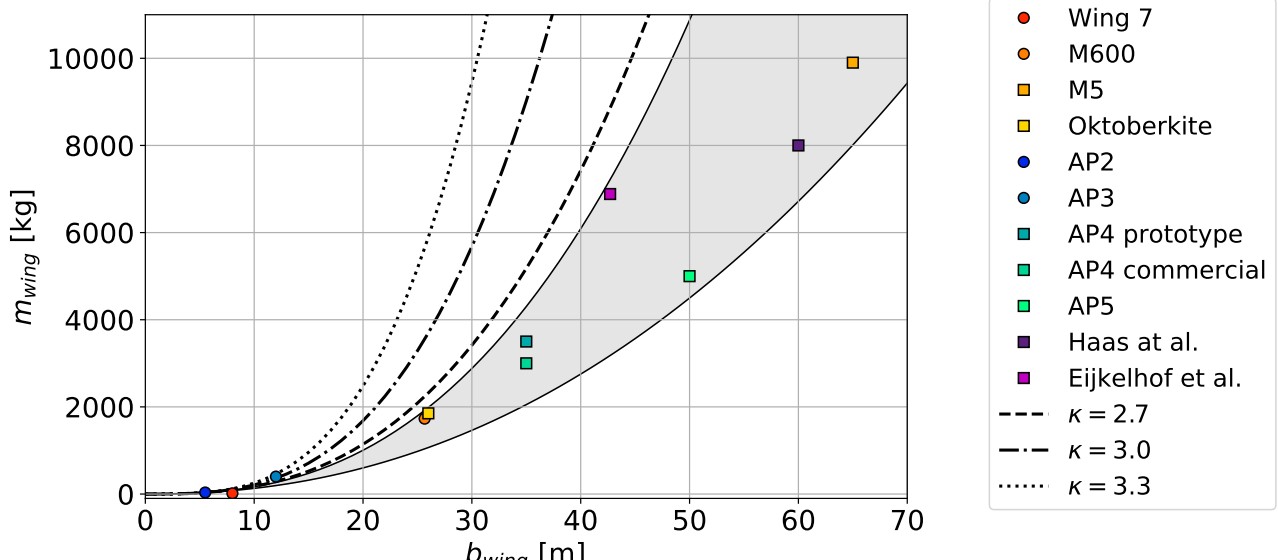

**Figure 3.** Published AWES aircraft masses (Haas et al., 2019; Kruijff and Ruiterkamp, 2018; Eijkelhof et al., 2020; Ampyx; Echeverri et al., 2020). Circular markers indicate built prototypes while square markers show estimated designs. For these data, the scaling exponent ranges between $\kappa = 2.2 - 2.6$ (grey area). The here investigated, more conservative mass scaling exponents between $\kappa = 2.7 - 3.3$, are depicted by dashed, dash-dotted and dotted lines.

## 3.5  Tether model

The tether is represented by multi-element, straight, cylindrical solid rods with constant diameter which cannot support compressive forces. This is a reasonable assumption when tether tension is high during the power production phase of the power cycle. The tether is divided into multiple 3 DOF tether nodes (here $n_{\mathrm{tether}} = 10$) that are connected by segments. The mass

assigned to each node is half of the mass of its connected tether segments, calculated with a constant material density of $\rho_{\text{tether}} = 970 \text{ kgm}^{-3}$. The tether node at the aircraft also contains the mass of the aircraft $m$. The drag contribution of each segment is equally divided between the its endpoints and propagated to either the aircraft or ground station. Assuming an even split between both nodes leads to an underestimation of total tether drag, especially when there is a large change in apparent wind speed along the tether length, because drag is proportional to the apparent wind speed squared (Leuthold et al., 2018; De Schutter et al., 2020, 2019). We assume a constant drag coefficient of $C_{\text{D}}^{\text{tether}} = 1$, which is the drag coefficient of a smooth cylindrical object at higher Reynolds numbers (typical for AWE applications) and could even be higher for braided tethers. The tether diameter $d$, and therefore drag, scales with tether tension and wing area, assuming constant tensile strength.

Tether force constraints are chosen such that the system's rated power is achieved at $U_{\text{sizing}}(100 \le z \le 400 \text{ m}) \approx 10 \text{ ms}^{-1}$, assuming a logarithmic wind speed profile, similar to wind at hub height for conventional wind turbines. Therefore, the tether diameter of every AWES design is derived from the maximum tether stress $\sigma_{\text{tether}} = 3.6 \cdot 10^9 \text{ Pa}$ and a safety factor $SF_{\text{tether}} = 3$.

The ground station is not explicitly modeled, instead hypothetical tether reeling speed and acceleration constraints are imposed based on previously publications (Licitra, 2018; Malz et al., 2019), mimicking rotational speed and motor torque limitations. Maximum reel-out speed is limited to $\dot{l}_{\text{in}} = 10 \text{ ms}^{-1}$ and reel-in speed to $\dot{l}_{\text{in}} = 15 \text{ ms}^{-1}$, resulting in a reel-out to reel-in ratio of $\frac{2}{3}$ which is assumed to be within design limitations of the winch. This limits the mechanical, instantaneous power that each ground-generation AWES can generate $P_{\text{inst.}}^{max} = F_{\text{tether}}^{\max} \dot{l}_{\text{out}}$. Tether acceleration $\ddot{l} = 10 \text{ ms}^{-2}$ and jerk $\dddot{l}_{\text{max}} = 100 \text{ ms}^{-3}$ are constraints to comply with generator torque limits.

### 3.6 Aerodynamic scaling

Figure 4 compares the aerodynamic performance of the AP2 wing with and without a 500 m tether to a high-lift wing. The solid lines show the aerodynamic coefficients of the untethered aircraft ($l = 0 \text{ m}$) and the dashed lines the ones of the tethered aircraft with a tether length of $l = 500 \text{ m}$. Lift $C_{\text{L}}$ (a), drag $C_{\text{D}}$ (b) and pitch moment $C_{\text{m}}$ coefficients (c) and glide ratio are depicted as functions of angle of attack (e). Lift-over-drag is shown in (d). Diagram (f) displays $\frac{C_{\text{R}}^3}{C_{\text{D}}^2}$ which determines the theoretical maximum power of any crosswind AWES as defined by Equation (2) (Loyd, 1980; Diehl, 2013). Echeverri et al. (2020) mention that two shortcomings of the original M600 design were the overestimation of $C_{\text{L}}^{\text{max}}$ and underestimation of $C_{\text{D}}$, prompting a more conservative estimation of practical aerodynamic coefficients. The aerodynamic coefficients of the AP2 reference model were identified by Licitra (2018); Malz et al. (2019) in AVL (Drela and Youngren) and confirmed through CFD analyses by Ampyx Power (Vimalakanthan et al., 2018) and during untethered test flights. Modifications to the AP2 aerodynamic reference model are implemented to assess the impact of improved aerodynamics on AWES performance (labeled HL for high-lift). This is achieved by shifting the $C_{\text{L}}$, $C_{\text{D}}$ and $C_{\text{m}}$ as if high-lift devices, such as fixed trailing-edge flaps and fixed leading-edge slots, were attached (Kermode et al., 2006; Lee and Su, 2011; Hurt, 1965; Scholz, 2016). This is achieved by increasing the lift, drag and moment and coefficients at $\alpha = 0$ and increasing the stall angle. The high-lift configuration does not represent a specific design, but an arbitrary improvement in aerodynamic efficiency, which is here defined as higher lift-to-drag ratio, in comparison to the reference AP2 data. Lift and drag at zero angle of attack are increased, stall is delayed, and pitch moment

decreased. (Loyd, 1980) The power harvesting factor $\zeta$ expresses the estimated AWES power $P$ relative to the total wind power through an area the same size as the wing $P_\text{area}$ and is defined as:

$$\zeta = \frac{P}{P_\text{area}} = \frac{\overline{P}}{\frac{1}{2}\rho_\text{air}AU(z)^3} \leq \frac{4}{27}c_\text{R}\left(\frac{c_\text{R}}{c_\text{D}}\right)^2 \tag{8}$$

It can be derived from (2) by setting the elevation angle $\varepsilon$ and the azimuth angle $\phi$ to zero. An extreme value analysis results in an optimal reel out speed $\dot{l}$ of 1/3 of the wind speed $U$ (Equation (1)) and $\zeta_\text{max} = \frac{4}{27}c_\text{R}\left(\frac{c_\text{R}}{c_\text{D}}\right)^2$. $U(z)$ is the wind speed and $\rho_\text{air}(z)$ the air density at operating altitude. While both airfoils have comparable optimal glide ratios (Figure 4 (e)), optimal $\zeta$ at zero elevation and azimuth angle (Figure 4 (f)) is more than twice as high for the high-lift airfoil ($\zeta_\text{HL}(l_\text{tether} = 500\text{ m}, \alpha = 7.23°) \approx 50$) compared to the AP airfoil ($\zeta_\text{AP2}(l_\text{tether} = 500\text{ m}, \alpha = 7.63°) \approx 23$).

Stall effects are implemented for both the AP2 reference model (blue) as well as the high-lift (HL - orange) model by fitting the lift curve to a quadratic function (Figure 4). As a result, the lift coefficients deviate slightly in the linear lift region at lower angle of attack.

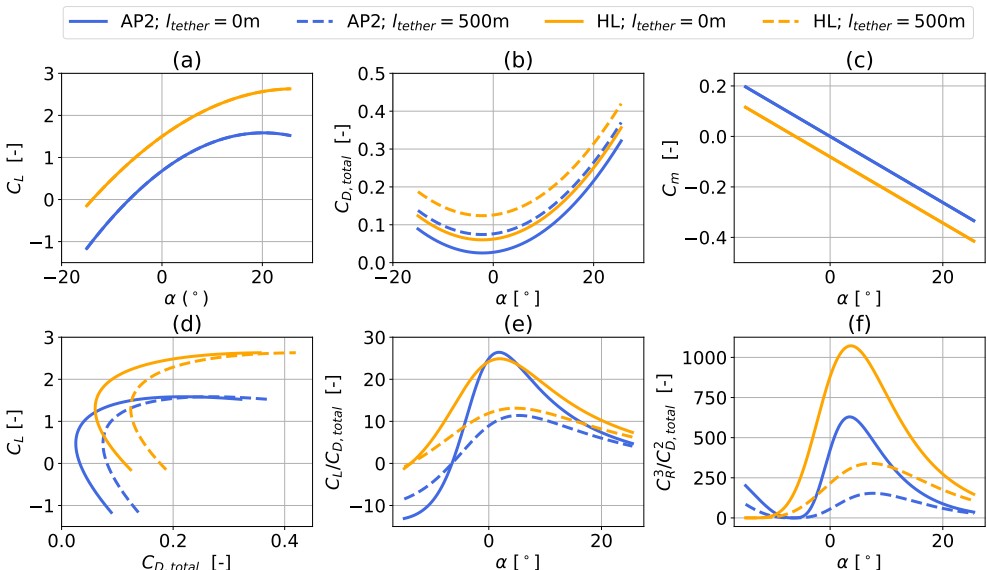

**Figure 4.** The dashed lines in (b) show $C_\text{D,total}$ (aircraft + 500 m tether) while the solid lines show $C_\text{D,wing}$ (only aircraft). Aerodynamic lift $C_\text{L}$ (a) and pitch moment $C_\text{m}$ coefficients (c), each with (dashed line) and without tether drag (solid line) as a function of angle of attack for AP2 (blue) (Licitra, 2018; Malz et al., 2019) and high-lift (HL) (orange) configuration. (d) displays lift as a function of total drag, (e) lift-to-drag ratio over angle of attack and (f) $c_\text{R}^3/c_\text{D,total}^2$ over angle of attack according to (Loyd, 1980). HL coefficients are derived by modifying the AP2 reference model as if arbitrary high-lift devices were attached.

## 3.7 Constraints

As previously mentioned, the AWES model solves a constrained optimal control problem to maximize average cycle power of a single 6 DOF tethered aircraft connected to the ground station via a straight inelastic tether. Each run optimizes the trajectory during the pumping cycle of an AWES at a fixed size for a given wind field (wind velocity as a function of altitude). Constraints include system dynamics, material properties, aircraft (Subsection 3.6) and ground station hardware as well as flight envelope limitations listed in Table 1. These limitations include minimum and maximum operating heights ($z_{\text{operation}}^{\min}$ and $z_{\text{operation}}^{\max}$), maximum acceleration $a_{\text{flight}}^{\max}$, measured as multiples of earths gravity, as well as a maximum tether length $l^{\max}$ to maintain safe operation. More information on the model and constraints can be found in De Schutter et al. (2020) and the therein referenced publications. The number of loops $n_{\text{loop}}$ within the reel-out phase of every pumping cycle is fixed to 5.

The maximum tether stress and force, from which the tether diameter is calculated, together with the periodicity constraint are some of the most important path constraints. Angle of attack $-30 \leq \alpha \leq 30$ and side slip angle $-15 \leq \beta \leq 15$ of the wing constraints ensure operation within realistic bounds. However, neither angular constraint is active during flight, because the optimizer tries to achieve an angle of attack close to the ideal harvesting factor $\zeta$ (Figure 4). Due to weight and drag effects, the actual angle of attack is closer to $\alpha \approx 10°$ during reel-out for the majority of wind speeds.

## 3.8 Initialization

The AWES dynamics are highly nonlinear and therefore result in a non-convex optimal control problem which possibly has multiple local optima. Therefore, the particular results generated by a numerical optimization solver can only guarantee local optimality, and usually depend on the chosen initialization. The optimization is initialized with a circular trajectory based on a fixed number of $n_{\text{loop}} = 5$ loops at a $30°$ elevation angle in positive $x$-direction and an estimated aircraft speed of $v_{\text{init}} = 10 \text{ ms}^{-1}$. Previous analyses showed that the convergence of large AWES strongly depends on initial tether length. Larger systems are less sensitive to tether drag, because lift-to-tether drag ratio scales linearly with wing area. Larger and heavier aircraft have a higher moment of inertia and hence have a larger turning radius, requiring a longer tether. Initial tether length $l_{\text{tether}}^{\text{init}}$ is increased linearly with aircraft wing area (Table 1) to improve the optimization process.

In order to solve the highly nonlinear optimization problem, an appropriate initial guess is generated using a homotopy method similar to those detailed in Gros et al. (2013); Malz et al. (2020). This technique gradually relaxes the problem from simple tracking of circular loops to the original nonlinear path optimization problem where the previous result serves as an initial guess for the following problem. An initial circular path, which is determined from the tether length guess and estimated flight speed, is transformed into a periodic helix-like trajectory. Several initial tether lengths were investigated to determine a feasible initial path depending on system mass, system size and wind speed. Initial tether lengths needs to increase with system size and wind speed. The resulting problem is formulated in the symbolic modeling framework CasADi for Python (Andersson et al., 2019, 2012) and solved using the NLP solver IPOPT (Wächter and Biegler, 2006) in combination with the linear solver MA57 (HSL).

Table 1. List of investigated AWES design parameters and selected system constraints for the six investigated designs with different wing sizes ($10\,\mathrm{m}^2 \leq A \leq 150\ \mathrm{m}^2$) with the original AP2 design as reference. The two different aerodynamic configurations (AP2 and HL) determine tether diameter $d$ and maximum tether force $F_{\mathrm{tether}}^{\max}$.

| | Parameter | AP2 | design 1 | design 2 | design 3 | design 4 | design 5 | design 6 |
|---|---|---|---|---|---|---|---|---|
| Aircraft | $A$ [m²] | 3 | 10 | 20 | 50 | 80 | 100 | 150 |
| | $c_{\mathrm{wing}}$ [m] | 0.55 | 1.00 | 1.41 | 2.24 | 2.83 | 3.16 | 3.87 |
| | $b_{\mathrm{wing}}$ [m] | 5.5 | 10 | 14.1 | 22.4 | 28.3 | 31.6 | 38.7 |
| | AR [-] | 10 | 10 | 10 | 10 | 10 | 10 | 10 |
| | $m_{\mathrm{aircraft}}(\kappa = 2.7)$ [kg] | 36.8 | 185 | 471 | 1,624 | 3,062 | 4,139 | 7,155 |
| | $m_{\mathrm{aircraft}}(\kappa = 3.0)$ [kg] | 36.8 | 221 | 626 | 2,473 | 5,005 | 6,995 | 12,850 |
| | $m_{\mathrm{aircraft}}(\kappa = 3.3)$ [kg] | 36.8 | 265 | 830 | 3,767 | 8,180 | 11,821 | 23,079 |
| | $\alpha$ [°] | [-10 : 30] | | | | | | |
| | $\beta$ [°] | [-15 : 15] | | | | | | |
| Tether | $l^{\max}$ [m] | 2000 | | | | | | |
| | $\dot{l}$ [ms$^{-2}$] | [-15 : 10] | | | | | | |
| | $\ddot{l}$ [ms$^{-2}$] | [-15 : 10] | | | | | | |
| | $\dddot{l}^{\max}$ [ms$^{-3}$] | [-20 : 20] | | | | | | |
| | $\rho^{\mathrm{tether}}$ [kgm$^{-3}$] | 970 | | | | | | |
| | $\sigma_{\max}^{\mathrm{tether}}$ [Pa] | $3.6 \cdot 10^9$ | | | | | | |
| | $SF^{sigma}$ [-] | 3 | | | | | | |
| | $d(\mathrm{AP2})$ [mm] | | 5.5 | 7.8 | 12.3 | 15.5 | 20 | 21.7 |
| | $d(\mathrm{HL})$ [mm] | | 7.2 | 10.2 | 16.1 | 20.6 | 23 | 28.3 |
| | $F_{\mathrm{tether}}^{\max}(\mathrm{AP2})$ [kN] | | 34 | 60 | 136 | 241 | 377 | 456 |
| | $F_{\mathrm{tether}}^{\max}(\mathrm{HL})$ [kN] | | 46 | 94 | 241 | 416 | 499 | 738 |
| flight envelope | $z_{\mathrm{operating}}^{\min}$ [m] | | 55 | 60 | 75 | 90 | 100 | 125 |
| | $z_{\mathrm{operating}}^{\max}$ [m] | 1000 | | | | | | |
| | $v_{\mathrm{flight}}^{\max}$ [ms$^{-1}$] | 80 | | | | | | |
| | $a^{\max}$ [ms$^{-2}$] | 12$\times$ g | | | | | | |
| | $n_{\mathrm{loop}}$ [-] | 5 | | | | | | |
| Initialization | $n_{\mathrm{loops}}$ | 5 | | | | | | |
| | $\varepsilon$ [°] | 30 | | | | | | |
| | $l^{\mathrm{init}}$ [m] | | 500 | 535 | 643 | 750 | 821 | 1000 |

## 4 Results and discussion

We compare six AWES sizes with three different mass properties and two sets of nonlinear aerodynamic coefficients each to investigate the design space and upscaling potential. Furthermore, we contrast the performance at representative onshore

(Pritzwalk in northern Germany) and offshore locations (FINO3 research platform in the North Sea) based on one year of WRF simulated and k-means clustered wind data. To that end, we show representative optimized trajectories (Subsection 4.1) and compare typical operating altitudes and tether lengths (Subsection 4.2). The dotted lines that connect the data points are only their to better visualize the data and do not indicate a smooth continuity between data points. Subsection 4.3 analyses AWES power curves for each design and determines power coefficient based on swept area and wing chord. From this we

derive the annual energy production (AEP) in Subsection 4.4 for each location and system configuration. We examine the predicted power losses (Subsection 4.6) due to tether drag. Finally, we establish an upper limit of the weight-to-lift ratio and compare tether drag forces in Subsection 4.5.

## 4.1 Flight trajectory and time series results

The trajectories in Figure 5 (b) and 5 (d) depict the local optima of the highly nonlinear model and optimization problem

for AWES designs 3 with a wing area of $A = 50 \text{ m}^2$, both reference (AP2, solid lines) and HL (dashed lines) aerodynamic coefficients and a scaling coefficient of $\kappa = 3$. The trajectories are within the set constraints and are consistent with other studies (De Schutter et al., 2019; Sommerfeld et al., 2020) which use the same model.

Figure 5 (a) shows the vertical wind speed profiles with the operating region highlighted in color. Any deviation from the WRF data in grey is caused by the interpolation with Lagrange polynomials during the implementation process described in

Subsection 3.3. The hodographs in Figure 5 (c) show a top view of the rotated horizontal wind velocity components $u$ and $v$ up to a height of 1000 m which follow the expected clockwise rotation with altitude (Stull, 1988).

Trajectories at higher wind speeds and above rated power deviate noticeably from the trajectories computed at lower wind speeds. The optimization algorithm tries to depower the aircraft by moving it out of the power zone of the wind window, which is the low elevation angle zone directly downwind. As a result, the trajectory either shifts upwards, increasing $\varepsilon$, or

295 sideways, increasing $\phi$, as can be seen from Equation (2), to stay within the tether force, tether reeling speed and flight speed constraints, while still maximizing average cycle power. Subsection 4.2 further analyzes the trend towards longer tethers and higher operating altitude with increasing wind speed.

Figure 6 shows the temporal evolution of the cycle trajectories depicted in Figures 5 (b) and 5 (d). The cycle duration varies with wind speed and system configuration. At lower wind speeds ($U_{\text{ref}} = 5.5 \text{ ms}^{-1}$), the better aerodynamics of the

300 HL configuration lead to higher flight speed and therefore shorter time to complete the cycle. At higher wind speeds the HL configuration needs to reduce the flight speed to stay within constraints and also requires a much longer reel-in phase, both leading to a longer cycle time. Because of the initialization with a simple circular trajectory and the fixed number of loop maneuvers, which are maintained during the optimization process. HL high wind speed optimizations show a loop maneuver during the reel-in period too. This is likely caused by the longer reel-in period required to return the aircraft to its initial position.

Because the tether tension is consistently high during reel-out (Figure 6 (a)), the reel-out speed remains high as well, leading to a longer reel-out length (Figure 6 (f)). As a result, the last loop is "carried over" to the reel-in period; for real deployed systems a modified trajectory would likely be adopted to deal with this condition.

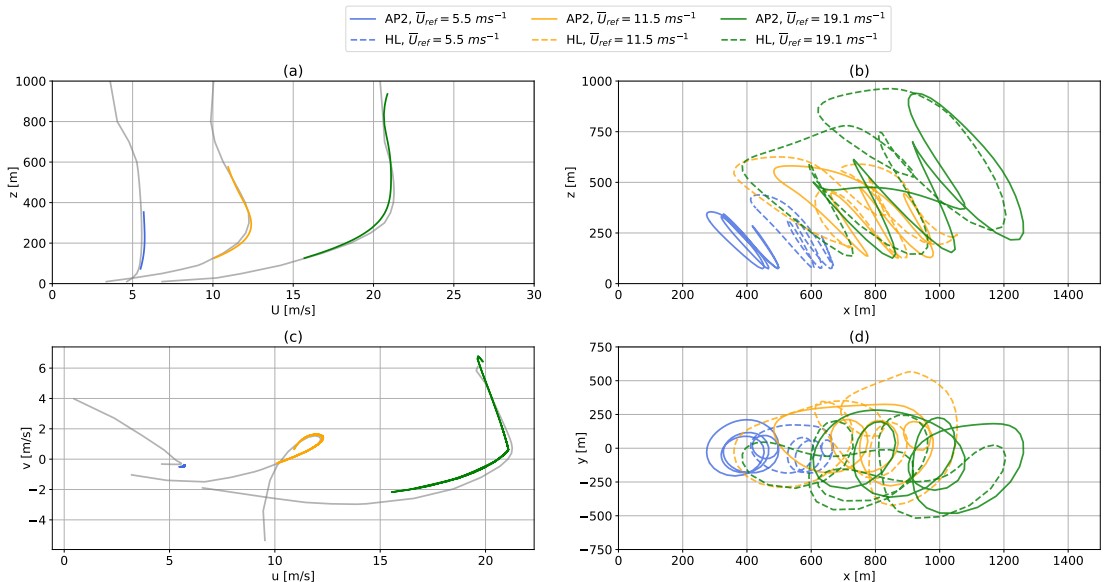

**Figure 5.** Optimization results of one pumping cycle for the ground-generation AWES with a wing area of $A = 50 \text{ m}^2$, mass scaling exponent $\kappa = 3$ for both AP2 reference (solid lines) and high-lift HL (dashed lines) aerodynamic coefficients at various WRF-generated wind conditions. Diagrams (a) and (c) depict representative horizontal onshore wind speed profiles and their hodographs of wind velocity up to 1000 m. The deviation of the colored lines is caused by the implementation of discrete WRF-simulated data points using Lagrange polynomials. Diagrams (b) and (d) show the optimized trajectories in side and top view.

Previous unpublished analyses utilizing the same model, showed that AWES power output seems to be fairly insensitive to both number of loops and flight time. This needs to be verified and compared to other models and real experiments.

The optimizer aims to achieve a constant, maximum tether force $F_{\text{tether}}$ (Figure 6 (a)) during the reel-out period and vary tether reel-out speed (Figure 6 (c)) to maximize power (Figure 6 (e)). This is achieved by varying the angle of attack (Figure 6 (d)), while trying to stay close to the optimal $C_{\text{R}}^3/C_{\text{D,total}}$ (Figure 4 (f)). At high wind speeds the angle of attack has to decrease to stay within tether tension constraints (orange and green lines). The trajectories are characterized by periodic cycles of aerodynamic forces and tether tension. In the production phase, the tether reels out and the aircraft follows an almost

circular pattern, which leads to deceleration of the aircraft during the ascent and acceleration during the descent, due to gravity. To maintain tether tension, tether reeling speed decreases to zero. At lower wind speeds the aircraft cannot produce sufficient lift force to pull the tether and overcome gravity during the ascent within each loop of the production cycle. As a result, tether force (Figure 6 (a)) decreases together with apparent wind speed $v_{\text{app}}$ (Figure 6 (b)), tether reeling speed $v_{\text{tether}}$ (Figure 6 (c)) and instantaneous power $P_{\text{inst}}$ ($U_{ref} = 5.5 \text{ ms}^{-1}$, blue) (Figure 6 (e)). Even at higher wind speeds ($U_{ref} = 11.5 \text{ ms}^{-1}$,

orange) the tether reeling speed drops to zero at during the ascent. As a consequence, the generated power also drops to zero and ramps up again (Figure 6 (e)), leading to grid feed-in challenges.

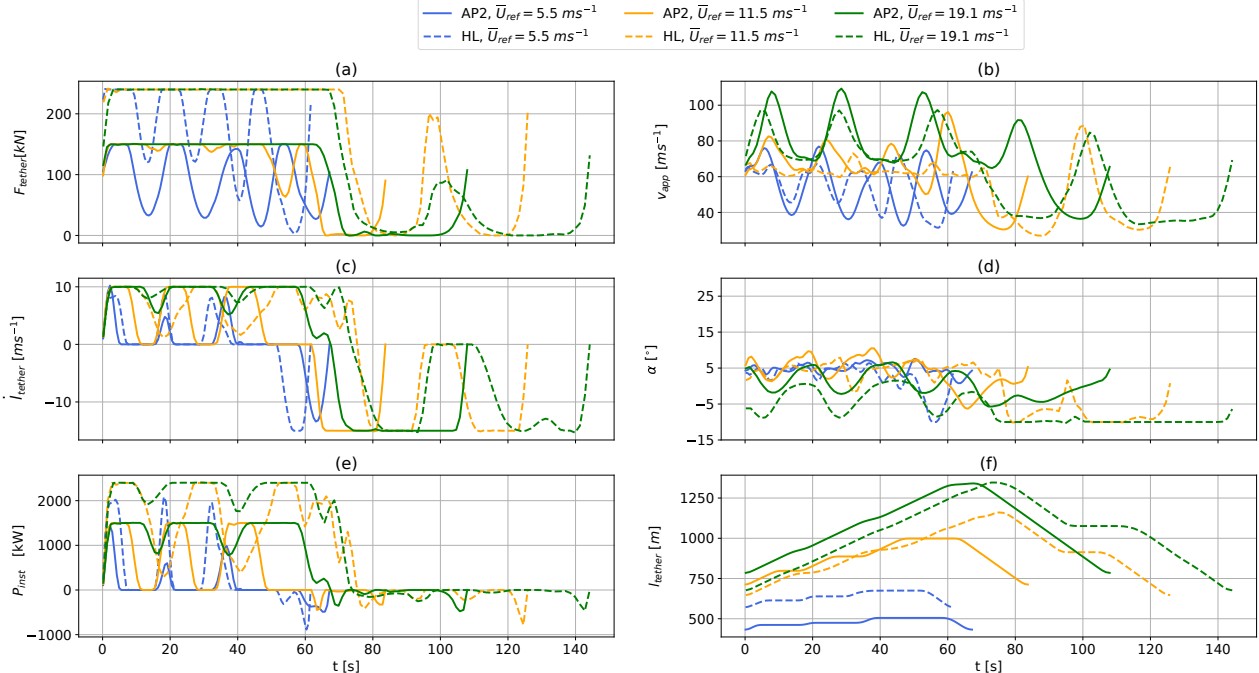

**Figure 6.** Time series of one optimized pumping cycle for the ground-generation AWES with a wing area of $A = 50$ m$^2$, mass scaling exponent $\kappa = 3$ for both AP2 reference (solid lines) and high-lift HL (dashed lines) aerodynamic coefficients at various WRF-generated wind conditions. The corresponding trajectories are shown in Figure 5. The diagrams show tether force $F_{\text{tether}}$ (a), apparent wind speed $v_{\text{app}}$ (b), tether reeling speed $v_{\text{tether}}$ (c), angle of attack $\alpha$ (d) and instantaneous power $P_{\text{inst}}$ (e), as well as tether length $l$ (f).

Additionally, aerodynamic loads drop to almost zero during the reel-in phase. To reduce the power losses and decrease the reel-in time, tether reeling speed quickly reaches its minimum of $v_{\text{tether}} = 15$ ms$^{-1}$. Buffering the energy or coupling multiple, phase-shifted AWES in a wind farm setup would be beneficial (Faggiani and Schmehl, 2018; Malz et al., 2018), to alleviate this inherent power fluctuation between the production (reel-out) and the consumption (reel-in) phases.

At lower wind speeds, the tether force (Figure 6, a blue line) approaches the maximum tether force during the peaks of the reel-out phase. With increasing wind speed, aerodynamic forces saturate due to tether tension constraints (Table 1), leading to increasing periods of constant, maximum tension. However, tether force troughs decrease even further with tether length, due to increased total system weight. Figure 7 gives an insight into the tether load cycles during the reel-out phases of AWESs with a wing area of $A = 50$ m$^2$. Average time between tether tension troughs $\Delta T(F_{\text{tether}}^{\min})$ (Figure 7 (a)) slightly increases with $\kappa$ due to increased aircraft inertia, but remains almost constant with wind speed $U_{\text{ref}}(100$ m $\leq z \leq 400$ m$)$. The relative reduction of tether tension troughs $\Delta F_{\text{tether}}/F_{\text{tether,max}} = |F_{\text{trough}} - F_{\text{tether,max}}|/F_{\text{tether,max}}$ decreases with wind speed as the apparent wind speed at the wing increases (Figure 7 (b)). Higher aerodynamic efficiency (HL circular marker and dotted line) increases performance and smooths out the troughs. Heavier systems with lower aerodynamic efficiency require a higher

wind speed to achieve constant tension during reel-out. Tether tension of the AP2 configuration at low wind speeds is not high enough to produce typical troughs, instead there are long periods of approximately zero tension, leading to missing data points. The lightest configuration achieves constant reel-out tension at around rated wind speed of $U_{\mathrm{ref}} = 10 \ \mathrm{ms}^{-1}$, which is the lowest wind speed at which the AWES can produce its rated power $\overline{P}_{\mathrm{rated}}$, while the heaviest design requires higher wind speeds of about $15 \ \mathrm{ms}^{-1}$.

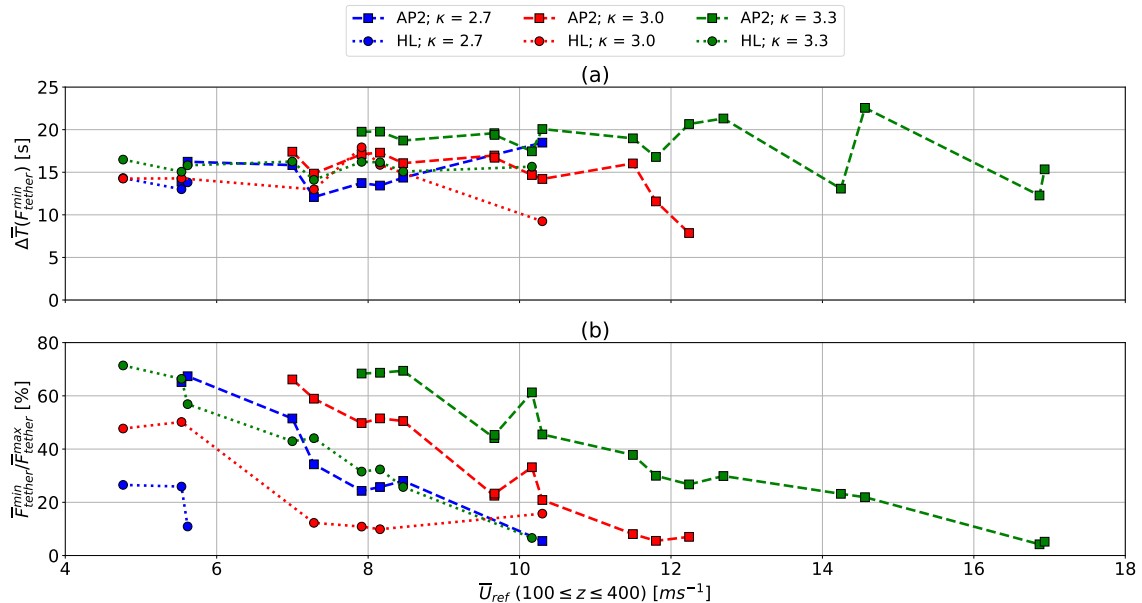

**Figure 7.** Average time $\Delta \overline{T}$ between tether tension troughs (a) and relative decrease in tether tension (b) during the production phase of optimized ground-generation AWES with a wing area of $A = 50 \ \mathrm{m}^2$. The diagram show results for mass scaling exponents $\kappa = 2.7, 3.0, 3.3$ (blue, red, green) and both sets of aerodynamic coefficients AP2 reference (square, dashed line) and high-lift HL (circle, dashed lines).

## 4.2 Tether length and operating altitude

One of the major value propositions of AWESs is that they can tap into wind resources beyond the reach of conventional wind turbines. The choice of optimal operating height strongly dependents on the vertical wind speed profile and system design. Two opposing effects influence the optimal operating height. On the one hand, an increase in altitude is generally associated with an increase in wind speed and therefore produced power. On the other hand, higher altitudes require a longer tether which
results in higher drag losses or an increase of the elevation angle which increase "cosine" losses (Diehl, 2013), or both.

Figure 8 shows a trend towards longer average tether lengths $\bar{l}_{\mathrm{tether}}$ (a), (b) and higher average operating altitudes $\overline{z}_{\mathrm{operating}}$ (c),(d) with increasing system size for a representative scaling exponent of $\kappa = 3$ (Equations (6) and (7)) and wind speed. Lighter aircraft and higher lift wings results in slightly higher operating altitudes, a longer tether and higher elevation angle.

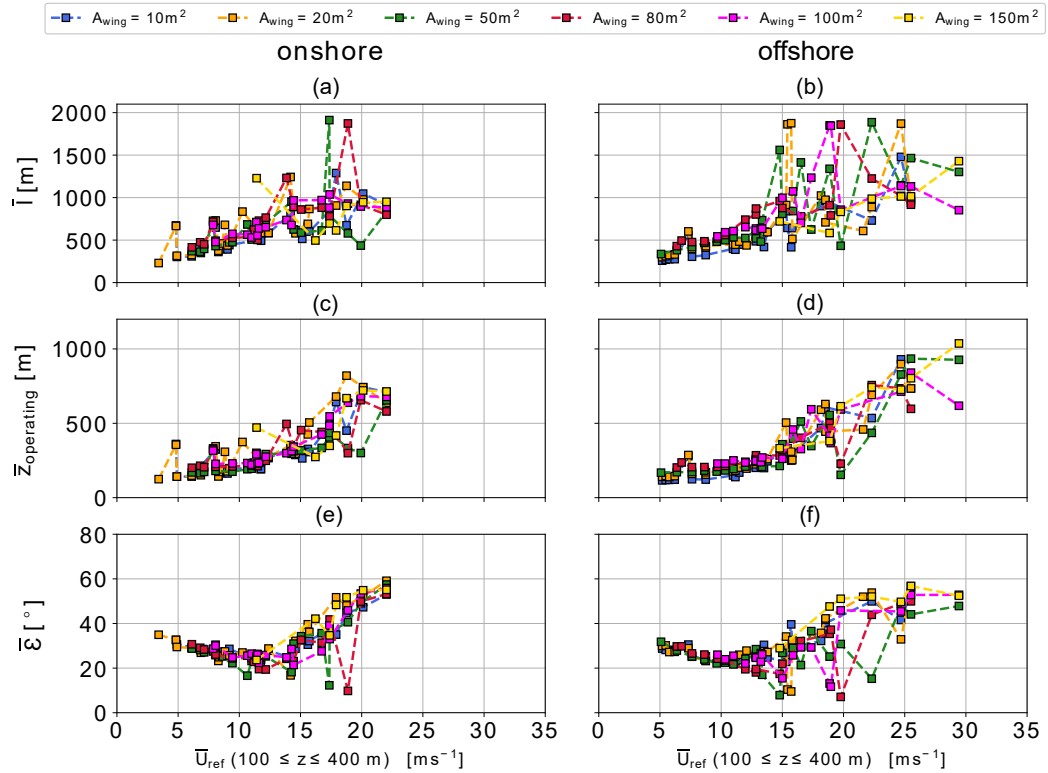

**Figure 8.** Average tether length $\bar{l}_{\text{tether}}$ (a), (b), average operating altitude $\bar{z}_{\text{operating}}$ (c),(d) and average elevation angle $\bar{\varepsilon}$ (e), (f) over reference wind speed $\overline{U}_{\text{ref}}(100 \leq z \leq 400 \text{ m})$. Results for wing areas between $A = 10 - 150 \text{ m}^2$ scaled with an exponent of $\kappa = 3$, AP2 reference aerodynamic coefficients for both onshore (left) and offshore (right) location.

Outliers, e.g. for high wind speed profiles (compare Figure 2), are likely local trajectory optima, which are still within the physical constraints of the highly nonlinear trajectory optimization problem described in Section 3. These outliers, which appear as non-monotonic variations in the plots, are the result of wind speed profile shape variations reflected in real profiles and the optimizer's solution trajectories for those profiles. Since we are not simply optimizing across a wind speed range with constant assumed shear profile, each wind speed solution is quite unique and potentially quite different even at similar nominal $\bar{U}_{r}ef$ value. This is in fact an important aspect highlighted by the current study, that AWES operations can be influenced much more than conventional wind turbines by the details of realistic wind profiles, and validates the impetus of our study in considering clusters of real wind profiles.

As wind speed increases beyond rated power ($U_{\text{ref}} \approx 10 \text{ ms}^{-1}$ Figures 5 and 6), the aircraft moves out of the wind window to depower. This is indicated by rising average elevation angles $\bar{\varepsilon}$ (bottom) above $U_{\text{ref}} = 10 \text{ ms}^{-1}$. Results for both offshore (right) and onshore (left) follow the same trends, but operating heights below rated wind speed are lower offshore because of lower wind shear and higher wind speeds.

It is important to keep in mind that even though the operating height exceeds 500 m for wind speeds of more than $U_{\text{ref}} \approx$ 15 ms$^{-1}$ such wind speeds occur only about 10 % of the time (Figure 2). These high wind speeds can significantly contribute to AEP. In case of the $A = 20$ m$^2$ wing, this contribution is about 29 % onshore and 33 % offshore. For $U_{\text{ref}}$ between 5 and 15 ms$^{-1}$, which is the most likely wind speed range, operating heights both onshore and offshore are between 200 to 300 m.

For smaller system sizes these heights are even lower. While this is slightly above the hub-height of current conventional wind turbines, it rebuts the argument of harvesting wind energy beyond this altitude (Archer and Caldeira, 2009). These findings are consistent with current offshore WT trends, whose rotor diameter increased significantly while hub height only increased marginally over the last years. It is likely that offshore hub heights will increase as technology improves, making the argument for the deployment of AWES particularity challenging as both operate at comparable heights and WT are the more proven and

established technology. However, this might be different for multiple kite systems which could benefit from longer tethers, due to reduced tether motion (De Schutter et al., 2019). Furthermore, the radical mass savings potential of AWES could prove a decisive factor to pursue the development.

## 4.3 Power curve, annual energy distribution and power harvesting factor

In the following, we compare average cycle power $\overline{P}$, annual energy production distribution $E$ and power harvesting factor $\zeta$

(Equation (8)) of optimized trajectories to the quasi steady-state model (QSM) described in Subsection 3.2.

Figures 9 (a) and 10 (a) compare the effect of aerodynamic efficiency and location on average cycle power in the form of a power curve for AWESs with a wing area of $A = 50$ m$^2$ and a mass scaling exponent of $\kappa = 2.7$. The data are derived from three representative profiles from each of the 10 wind velocity clusters. the average wind speed between 100 and 400 m has been chosen as reference wind speed, because these are typical operating heights for AWES. Using this altitude range results

in comparable power curve trends onshore and offshore. Offshore AWES could benefit from a larger tether diameter as wind speeds are generally higher (Figure 2) which would result in higher rated power. Higher lift coefficients result in higher rated power and a steeper power increase up to rated power. Power variations are caused by local optima mostly occurring above rated wind speed as the system depowers to stay within tether force and flight speed constraints (Subsection 3.7).

Rated power $\overline{P}_{\text{rated}}$, defined as the maximum generated power, which is constrained by instantaneous tether force and reeling

speed, is summarized in Table 2. Tether reel-in and reel-out speed constraints are kept constant for all designs, simulating drum speed constraints. Tether diameter is kept constant for both locations, but adjusted according to aircraft wing area and aerodynamic efficiency so that all system sizes reach rated power at about $U_{\text{ref}} = 10$ ms$^{-1}$ (Subsection 3.5). Therefore, the HL configuration achieves higher rated power. No cut-out wind speed limitations are implemented. Therefore, wind power is only limited by each location's maximum wind speed of the data set, which is significantly higher offshore (compare Figure 2).

Table 2 also shows the estimated equivalent WT rotor diameter $D_{\text{WT}}^{\text{equiv}}$, for an assumed power coefficient of $c_{\text{p}}^{WT} = 0.4$ and a rated wind speed of 10 ms$^{-1}$. The system size and therefore material cost benefits of AWES become obvious when comparing AWES wing span $b_{\text{wing}}$ to WT rotor diameter $D_{\text{WT}}^{\text{equiv}}$. AWES wing span is about 30 (HL) to 40 % (AP2) of the equivalent rotor diameter.

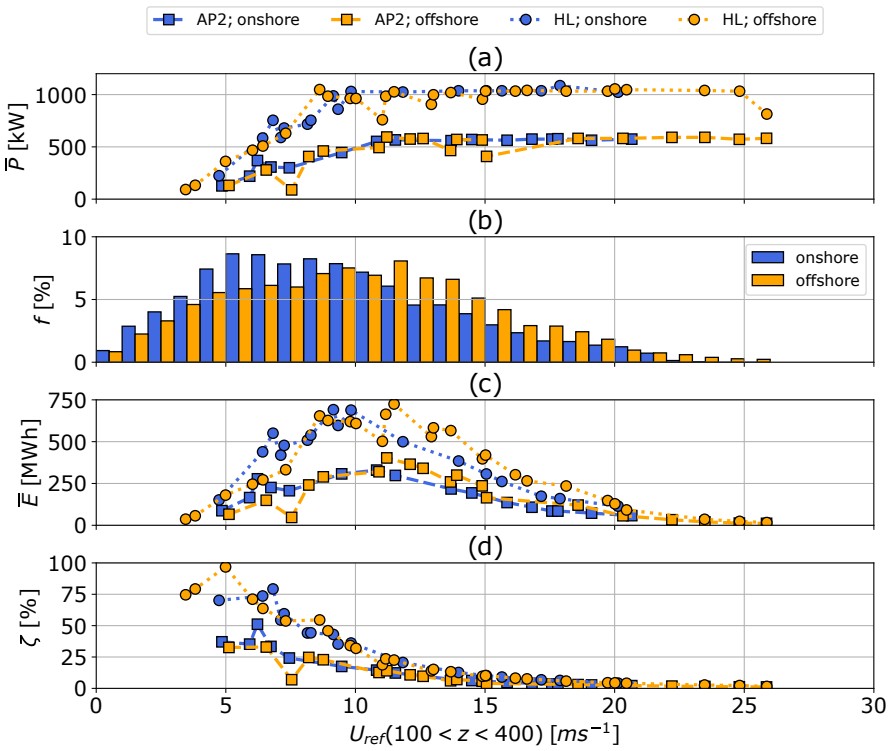

**Figure 9.** Representative power curves (a) for both sets of HL (circle) and AP2 (square) reference aerodynamic coefficients for both onshore (blue) and offshore (orange) location. The masses of the $A = 50$ m$^2$ wing area aircraft are scaled according to Equations (6) and (7) with a mass exponent of $\kappa = 2.7$. Average cycle power $\overline{P}$ is derived from p5, p50, p95 wind velocity profiles within each of the $k = 10$ WRF-simulated clusters. Diagram (b) presents the average annual wind speed probability distribution over reference height range of $100 \leq z_{\mathrm{ref}} \leq 400$ m. The annual energy production distributions over the wind speed are depicted in (c). Diagram (d) shows the corresponding harvesting factor $\zeta$.

**Table 2.** Rated power (cycle-average) of AWES with a mass scaling exponent of $\kappa = 2.7$ and equivalent wind turbine rotor diameter

| $A[\mathrm{m}^2]$ | 10 | | 20 | | 50 | | 80 | | 100 | | 150 | |
|---|---|---|---|---|---|---|---|---|---|---|---|---|
| $b_{\mathrm{wing}}[\mathrm{m}]$ | 10 | | 14.1 | | 22.4 | | 28.3 | | 31.6 | | 38.7 | |
| aerodynamic coeff. | AP2 | HL | AP2 | HL | AP2 | HL | AP2 | HL | AP2 | HL | AP2 | HL |
| $\overline{P}_{\mathrm{rated}}$ [kW] | 145 | 200 | 265 | 420 | 575 | 1030 | 1045 | 1800 | 1600 | 2225 | 2000 | 3400 |
| $D_{\mathrm{WT}}^{\mathrm{equiv}}$ [m] | 27 | 32 | 37 | 47 | 55 | 73 | 74 | 97 | 91 | 108 | 102 | 132 |

AEP and cf almost doubles for HL in comparison to the AP2 reference, highlighting the importance of exploring high-lift

configurations. The QSM modeled power curves (Figure 10), which use the same wind velocity profiles and tether diameter as the optimization model, achieve rated power at around $U_{\mathrm{rated}}(100 < z_{\mathrm{ref}} < 400) \approx 8$ ms$^{-1}$. This is caused by the fact that the engineering model neglects mass and predicts optimal power production, whereas the dynamic optimization model resolves the

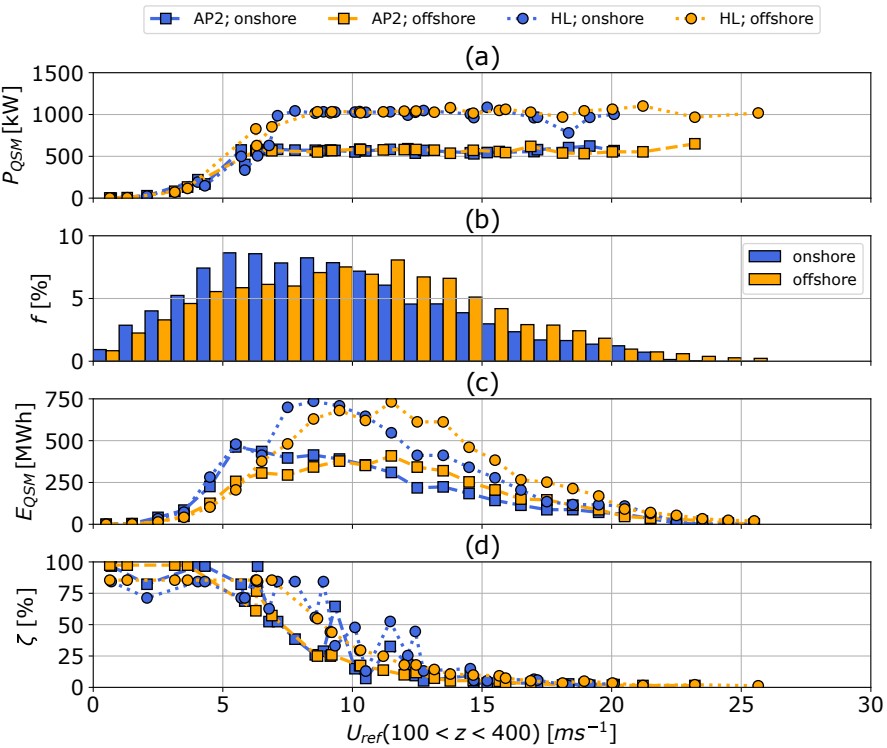

**Figure 10.** QSM based power curves (a) for a wing area of $A = 50$ m$^2$, both sets of HL (circle) and AP2 (square) reference aerodynamic coefficients and both onshore (blue) and offshore (orange) location. Optimal power $\overline{P}_{\mathrm{Loyd}}$ is derived from p5, p25, p50, p75, p95 wind speed profiles within each of the $k = 10$ WRF-simulated clusters. Diagram (b) presents the average annual wind speed probability distribution over reference height range of $100 \leq z_{\mathrm{ref}} \leq 400$ m. The annual energy production distributions over the wind speed are depicted in (c). Diagram (d) shows the corresponding harvesting factor $\zeta$.

flight trajectory and the varying forces and power within each production cycle. Deviation between QSM onshore and offshore power are due to variation in wind conditions.

The annual wind speed probability distributions $f$ in Figures 9 and 10 (b) represent the average annual wind speed between $100 \leq z \leq 400$ m which stands in as a proxy for wind at operating altitude (Section 2). As expected, higher wind speeds are more likely to occur offshore (FINO3) than onshore (Pritzwalk). Very high wind speeds above $U_{\mathrm{ref}} > 18 - 20$ ms$^{-1}$, beyond the cut-off speed of realistic wind energy converters, have a very low occurrence at both locations. The resulting annual average energy production distributions $\overline{E}$ (Figures 9 (c) and 10 (c)) reveal a clear difference between the offshore and onshore energy

potentials.

Estimated energy production distributions $E_{\mathrm{QSM}}$ (Figure 10 (c)) of the QSM reference model are based on the same wind speed probability distribution as the optimization model. Here the QSM data has been interpolated to be compatible with the annual wind speed probability distribution $f$ (Figure 10 (b)). The QSM predicts a higher energy production distribution (Figure

10 (c)) than the optimization model up to rated wind speed, because of the lack of a defined cut-in wind speed. Beyond rated

power, $E_{\mathrm{QSM}}$ is similar to optimized results, as predicted power is very similar, except some small variation. This leads to a higher AEP and cf predictions (Figure 11).

Figures 9 (d) and 10 (d) present the power harvesting factor $\zeta$ Diehl (2013), which sets cycle-average power $\overline{P}$ in relation to the total wind power of a cross sectional area $P_{\mathrm{area}}$ of the same size as a given wing $A$. The power harvesting factor decreases steadily for both the optimization and QSM. The QSM predicts an almost constant $\zeta$ at low wind speeds ($U_{\mathrm{ref}} < 5 \mathrm{~ms}^{-1}$).

## 4.4 Annual energy production and capacity factor

The previously described power curves $\overline{P}$ (Figures 9 and 10(a)) and annual wind speed probability distributions $f$ (Figures 9 and 10(b)) allow the investigation of the annual energy production distribution $E$ (Figures 9 and 10(c)). AEP is derived from the binned average cycle power $P_{\mathrm{i}}$, its corresponding wind speed probability $f_i$ and the total hours per year.

$$\mathrm{AEP} = \sum_{\mathrm{i}=1}^{\mathrm{k}} \left( \overline{P}_{\mathrm{i}} f_{\mathrm{i}} \right) \ 8760 \ \frac{\mathrm{h}}{\mathrm{year}} \tag{9}$$

cf is calculated from AEP and rated system power $P_{\mathrm{rated}}$ (Table 2), defined as the maximum average cycle-power.

$$\mathrm{cf} = \frac{\mathrm{AEP}}{P_{\mathrm{rated}} 8760 \ \frac{\mathrm{h}}{\mathrm{year}}} \tag{10}$$

We assume the same wind speed probability distribution for the QSM model as for the optimization model. Figure 11 compares AEP for all system sizes scaled with a mass scaling exponent of $\kappa = 2.7$ to QSM data. Figures 11 (a) and 11 (c) describe onshore conditions, while Figures 11 (b) and 11 (d) describe offshore conditions. AEP increases almost linearly with

425 wing area, because power scales linearly with wing area when keeping the maximum tether reeling speed constant throughout all optimization runs. As expected, HL aerodynamic coefficients (circle) outperform the AP2 reference (square). Offshore (orange) AEP and cf is generally higher than onshore (blue) because of the higher likelihood of higher wind speeds. The QSM predicts higher AEP, because of the previously described differences in power up to rated wind speed (Subsection 4.3), but follows the same trends. The optimization model predicts lower average AEP at $A = 150 \mathrm{~m}^2$, due to the high number

of infeasible solutions at lower wind speeds. Overall cf (Figure 11 c,d) remains almost unchanged up to $A = 100 \mathrm{~m}^2$ and sharply declines for $A = 150 \mathrm{~m}^2$. Onshore AEP and cf seems to outperform offshore for wing areas larger than $100 \mathrm{~m}^2$. This is likely caused by outliers, or wind velocity profile specific local minima, in the power curve before rated wind speed ($v_{\mathrm{rated}} = 10 \mathrm{~ms}^{-1}$), where the system seemingly over-performs. The QSM predicts very high cf values at both locations, while offshore AEP always outperforms onshore AEP. The relatively high cf values are the result of relativity low rated wind speed.

This location-specific design trade-off between generator size, wing area and tether diameter needs to be further investigated.

Figure 12 compares AEP for a mass scaling exponents of $\kappa = 2.7$ to scaling with $\kappa = 3$ and $\kappa = 3.3$, both onshore and offshore. High mass configurations with no feasible trajectory at any wind speed, which could be interpreted as the wind speed

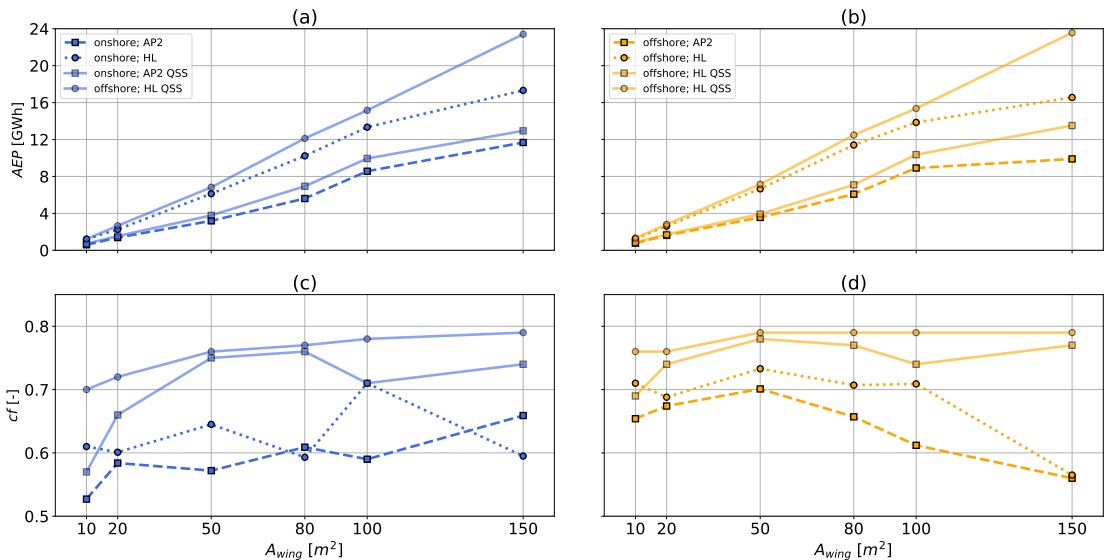

**Figure 11.** Representative AEP (a,b) and cf (c,d) as a function of aircraft wing area $A$ scaled according to Equations (6) and (7) with a mass exponent of $\kappa = 2.7$. QSM (solid lines) results are included for reference (Subsection 3.2). The Diagrams summarizes data for both sets of HL (circle) and AP2 (square) aerodynamic coefficients as well as both onshore (left, blue) and offshore (right, orange) location. Results are based on the average cycle power $\overline{P}$ derived from p5, p50, p95 wind velocity profiles within each of the $k = 10$ WRF-simulated clusters and wind speed probability distribution between $100 \leq z_{\mathrm{ref}} \leq 400$ m, used as a proxy for wind speed at operating height.

is below the cut-in wind speed, result in missing data. While smaller systems seem to be almost unaffected by aircraft weight, mass scaling effects lead to significant reduction in AEP for larger AWES. This is particularly true for wings with the AP2

reference aerodynamic coefficients (AP2, square) and onshore wind conditions. Combining results from both Figure 11, which already shows diminishing returns in AEP and cf with increasing wing area for the lightest, idealized aircraft mass scaling, and Figure 12, which predicts that AEP will only decline for heavier mass scaling, conveys that upscaling AWES is only beneficial with significant weight reduction. These results hint at the existence of an upper limit of fixed-wing AWES weight relative to AWES size or lift (Subsection 4.5), which is plausible since mass scales with aircraft volume, assuming pure geometric scaling

according to the square-cube law, and lift scales with aircraft area. The mass of soft wing AWES, which are hollow tensile structures filled with air, scales to a great extent with the wing surface, leading to significantly lower mass scaling exponents and more beneficial mass scaling. To account for these scaling effects and considering the power fluctuation caused by the cyclic nature of ground-generation AWES, it is likely better to deploy multiple smaller scale devices rather than a single large-scale system. The ideal, site-specific system size needs to be determined by realistic, achievable mass scaling and the local

wind resource.

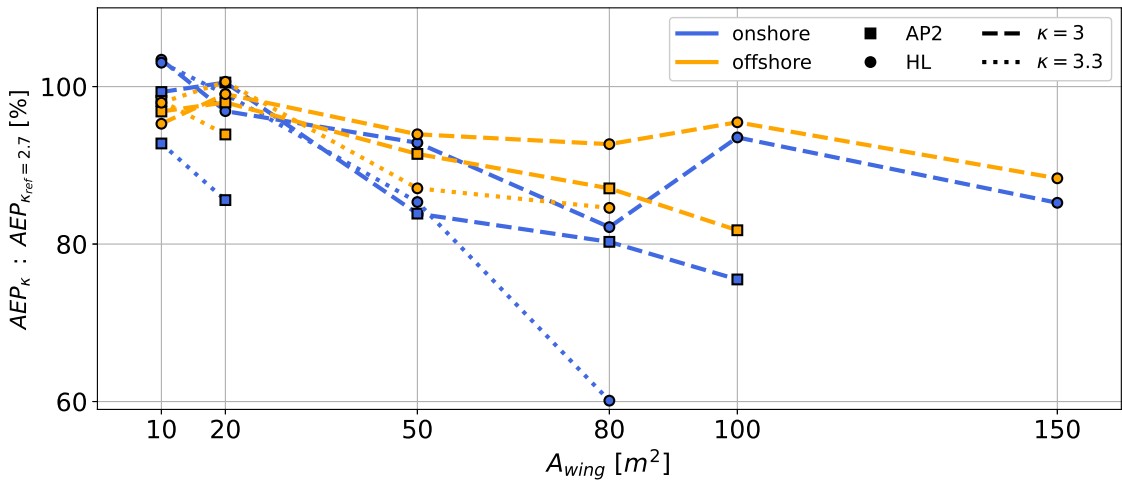

**Figure 12.** AEP ratio for mass scaling exponent $\kappa = 3$ (dashed lines) and $\kappa = 3.3$ (dotted lines) relative to AEP of $\kappa = 2.7$ as a function of aircraft wing area $A$. The diagram summarizes data for both onshore (blue) and offshore (orange) locations as well as both sets of aerodynamic coefficients HL (circle) and AP2 (square). Results are based on the average cycle power $\overline{P}$ derived from p5, p50, p95 wind velocity profiles within each of the $k = 10$ WRF-simulated clusters. Missing data points indicate that no feasible solution for any wind velocity profile was found.

## 4.5 Impact of weight and drag

The crosswind AWES concept exploits the increased apparent wind speed generated by the flight motion of the tethered aircraft (Loyd, 1980). Such trajectories, whether circular or figure-of-eight, always include an ascent during every loop maneuver where the aircraft needs to overcome gravity to gain altitude. This leads to a deceleration and therefore reduction of aerodynamic lift.
AWESs with excess mass fail to overcome weight and drag and can no longer climb during these phases.

With an increased wing area, the entire aircraft, particularly the load-carrying structures such as the wing box, need to increase in size and weight in order to withstand the increased aerodynamic loads produced by a larger wing. When tether drag is considered power scales faster than $b^2$, because tether drag losses are proportional to the tether diameter, which scales relative to the square root of the wing area. Similarly, conventional WT power and AEP scales with the rotor diameter square,
while theoretic WT mass scales with the cube of the rotor diameter. Comparing both wind energy converters under these assumptions, AWES perform worse with size as their flight path degrades. This can be attributed to the fact that AWES need to produce enough lift to carry their own weight to maintain operational, while WT are supported by a tower.

These facts limit AWES size, as the prevailing wind resource does not improve enough to produce sufficient aerodynamic lift to overcome the increased system drag and weight. An increase of operating altitude only comes with a marginal wind speed
increase especially offshore (compare Figure 2). Furthermore, higher operating altitudes also lead to increased cosine losses,

unless offset by a longer tether which in turn results in more drag and weight. Better aerodynamics or lighter, more durable aircraft and tether materials can only push this boundary, but not overcome it.

A comparison of tether weight $W_{\text{tether}}$ to total system weight during the production phase (reel-out) in Figure 13 (a) shows that the tether on average makes up 10 to 30 % of the entire airborne system weight. To give a general overview of these trends,
these figures show the averaged weight and drag over the entire wind speed range. Note that the tether cross sectional area is sized with a safety factor of 3. Tether cross sectional area mostly scales with aerodynamic force and therefore wing area while the aircraft weight scales with a mass scaling exponent $\kappa$ which results in decreasing trend lines. This value is higher for high-lift airfoils (circle) as the tether diameter is larger to withstand higher aerodynamic forces. For lighter aircraft, scaled with $\kappa = 2.7$ (dash-dotted), the portion of tether weight is higher because the tether diameter remains constant while the aircraft
mass is lighter.

Figure 13 (b) reveals that tether drag makes up about 18 to 40 % of the entire airborne system drag during the production phase. Tether diameter $d$ and therefore drag area ($A_{\text{tether}}^{\text{drag}} = d\, l$) scales beneficially with wing area, leading to the downward trend lines.

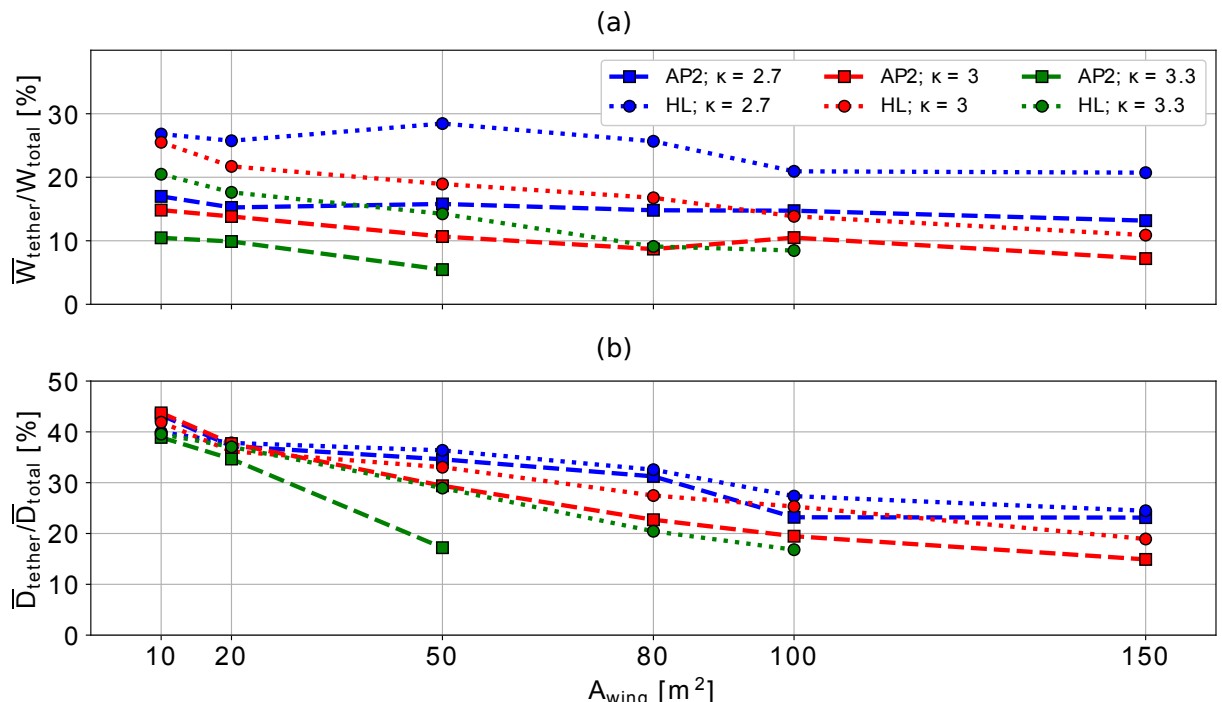

**Figure 13.** Percentage of cycle-average tether weight $\overline{W}_{\text{tether}}$ to total weight of airborne components $\overline{W}_{\text{total}}$ (a) and tether drag $\overline{D}_{\text{tether}}$ to total drag $\overline{D}_{\text{total}}$ (b) during production phase (reel-out), averaged over the entire wind speed range, for all aircraft sizes $A = 10 - 150\,\text{m}^2$, sets of aerodynamic coefficients AP2, HL and mass scaling exponents $\kappa = 2.7, 3, 3.3$ for wind data at the offshore location.

It is critical for crosswind AWES to ascend during each loop of the production or reel-out phase. The aircraft increases angle of attack (Figure 6) to compensate for the decreased apparent wind speed. For larger and heavier systems, this is not enough to maintain aerodynamic force and tether tension during times of lower wind speeds. Figure 14 (a) shows the aeronautic load factor during the production phase. It is defined as the ratio of average pattern trajectory lift force $\overline{L}_{\mathrm{wing}}$ to total AWES weight $\overline{W}_{\mathrm{total}}$, including tether which is sized with a safety factor of 3 and aircraft. The average load factor decreases from about 10 - 20 to 5 - 10, depending on aerodynamic performance and mass scaling, which is approximately the maneuvering load factor of an acrobatic airplane $n_{\mathrm{acrobatic}} = 6.0$ (Federal Aviation Agency, 2017). For utility airplanes this value is about $n_{\mathrm{acrobatic}} = 4.4$. The AWES aeronautic load factor is relatively high in comparison to untethered aircraft, because the high lift coefficient, which are designed to maximize traction power, in combination with the high wind speeds during the crosswind motion, lead to very high lift forces. The beneficial effect of better aerodynamics and mass scaling are clearly visible in a higher lift-to-weight ratio. High system mass with insufficient lift on the other hand leads to infeasible solutions and missing data.

Figure 14 (b) shows a slight reduction of total average drag $\overline{D}_{\mathrm{total}}$ to average lift $\overline{L}_{\mathrm{wing}}$ ratio with increasing wing area. Overall, this ratio remains almost constant between 6 to 8 %. The increase for $A = 100, 150 \mathrm{~m}^2$, $\kappa = 3$ and AP2 aerodynamics is likely caused by local optimization minima and few feasible wind speed profiles.

For a large-scale aircraft with an area of $A = 150 \mathrm{~m}^2$, scaled with the lightest mass scaling exponent of $\kappa = 2.7$, and AP2 reference aerodynamic coefficients, no feasible solution could be found for low wind speeds $U_{\mathrm{ref}} < 5 \mathrm{~ms}^{-1}$. This can be seen in Figure 15 which shows the average lift $\overline{L}_{\mathrm{wing}}$ divided by total weight $\overline{W}_{\mathrm{total}}$, including tether and aircraft, for all aircraft sizes with AP2 reference aerodynamic scaled with $\kappa = 2.7$. weight-to-lift ratio increases up to $U_{\mathrm{ref}} \approx 5 \mathrm{~ms}^{-1}$, above which it remains almost constant. Deviation from the expected trend lines are likely caused by the wind speed profile shapes and variations in optimized trajectory, especially towards higher wind speeds. To stay within the constraints, the optimizer determines trajectories which vary in shape, operating altitude and tether length from the typical trajectories found at lower wind speeds. This can likely be attributed to the applied apparent flight speed constraint of $U_{\mathrm{app}}^{\mathrm{max}} = 80 \mathrm{~ms}^{-1}$ which seems to already be achieved at this reference wind speed.

From this, together with time series data shown in Figure 6, it is possible to estimate the minimum cut-in wind speed or minimum viable aerodynamic load factor (lift to weight ratio). For the investigated design and constraints, the minimum viable aerodynamic load factor seems to be about 5 which is equivalent to a maximum viable weight-to-lift ratio of 20 %. No feasible solutions were found for lower wind speeds.

Figure 15 (b) shows the lift $\overline{L}_{\mathrm{wing}}$ to total drag $\overline{D}_{\mathrm{total}}$, including tether drag, ratio over reference wind speed for all aircraft sizes scaled with $\kappa = 2.7$ and AP2 reference aerodynamic coefficients. Data for all aircraft sizes show a similar trend. As tether length increases with wind speed (Figure 8), the total system drag and weight increases as well. Subsequent changes to angle of attack $\alpha$ raise lift force which increases the lift to drag ratio.

## 4.6  Power losses

An increased aircraft wing area not only leads to increased power potential, but is also accompanied by increased tether losses due to weight and drag. Tether mass scales with aircraft wing size because the higher aerodynamic forces require a larger tether

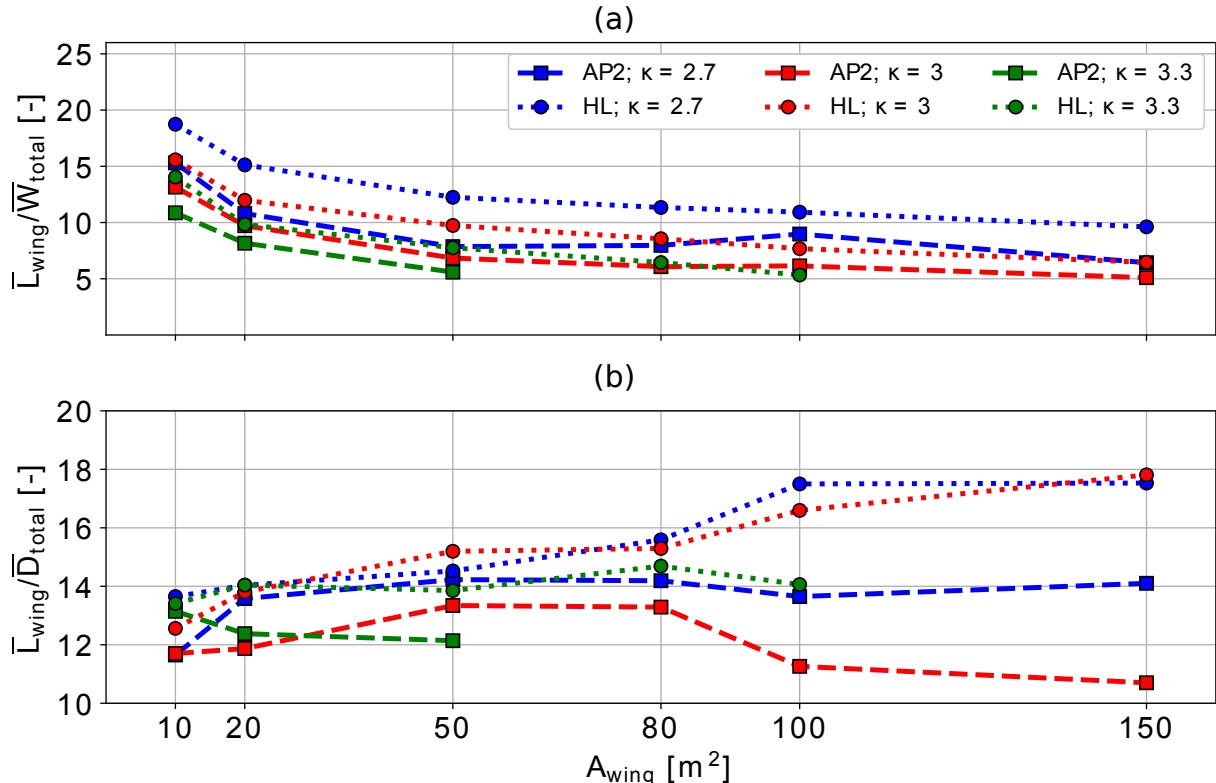

**Figure 14.** Load factor or lift $\overline{L}_{\mathrm{wing}}$ to $\overline{W}_{\mathrm{total}}$ ratio (a) and cycle-average total lift $\overline{L}_{\mathrm{wing}}$ to drag $\overline{D}_{\mathrm{total}}$, including tether drag, (b) during production phase (reel-out) for all aircraft sizes $A = 10 - 150 \ \mathrm{m}^2$, sets of aerodynamic coefficients AP2, HL and mass scaling exponents $\kappa = 2.7, 3, 3.3$ for wind data at the offshore location. Large-scale results for $A = 100, 150 \ \mathrm{m}^2$ might be misleading because only high wind speeds result in feasible solutions (compare figures 15).

diameter, assuming constant tether strength. Tether length increases with AWES size and wind speed (Subsection 4.2) which further increases tether drag and weight.

Figure 16 compares the average power losses due to the tether $\overline{P}_{\mathrm{tether}}^{\mathrm{drag}}$, calculated from the tether drag assigned to the aircraft node and its flight speed, relative to average cycle power $\overline{P}$. This power loss can be interpreted as how much of the harvested wind power is dissipated by the tether. Indirect power losses associated with a larger tether, such as the reduction in flight speed due to drag and weight, are not included in this analysis. The relative tether drag loss decreases with wing area, because tether diameter scales beneficially with the square root of the tether force which scales linearly with wing area. This scaling trend is

encouraging, but is counteracted and dominated by mass increases with size highlighted in earlier sections. As expected, the high-lift airfoil HL (dotted lines) experiences less relative drag loss than the AP2 reference airfoil (dashed lines) due to higher average cycle power.

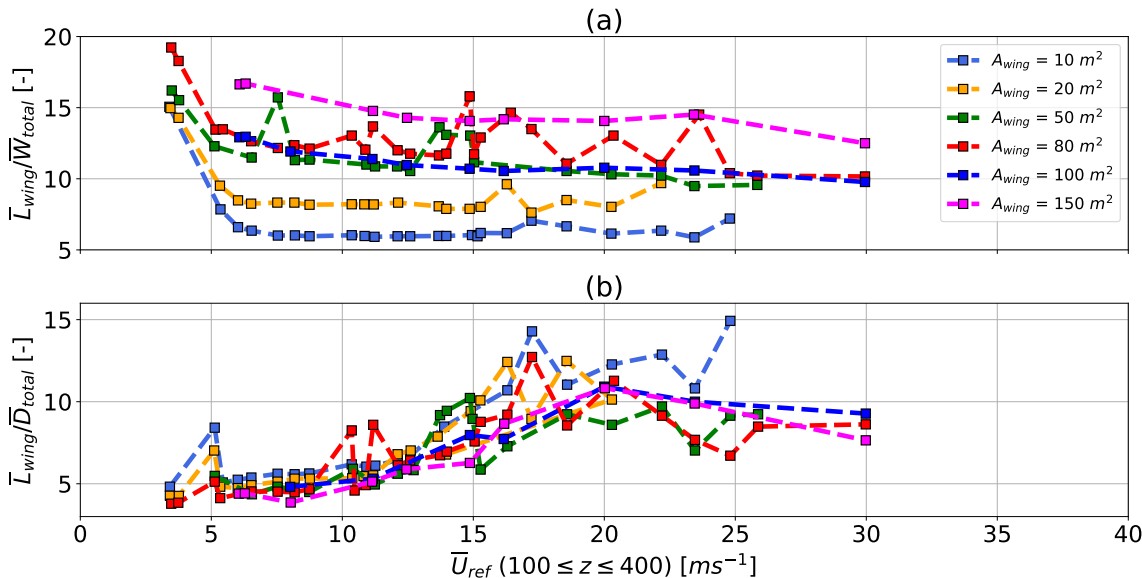

**Figure 15.** Ratio of cycle-average total weight $\overline{W}_{\text{total}}$ to lift $\overline{L}_{\text{wing}}$ (a) and cycle-average total drag $\overline{D}_{\text{total}}$, including tether drag, to lift $\overline{L}_{\text{wing}}$ (b) during production phase (reel-out) for all aircraft sizes $A = 10 - 150\,\text{m}^2$ for AP2 reference aerodynamic coefficients and a mass scaling exponent of $\kappa = 2.7$ over reference wind speed offshore.

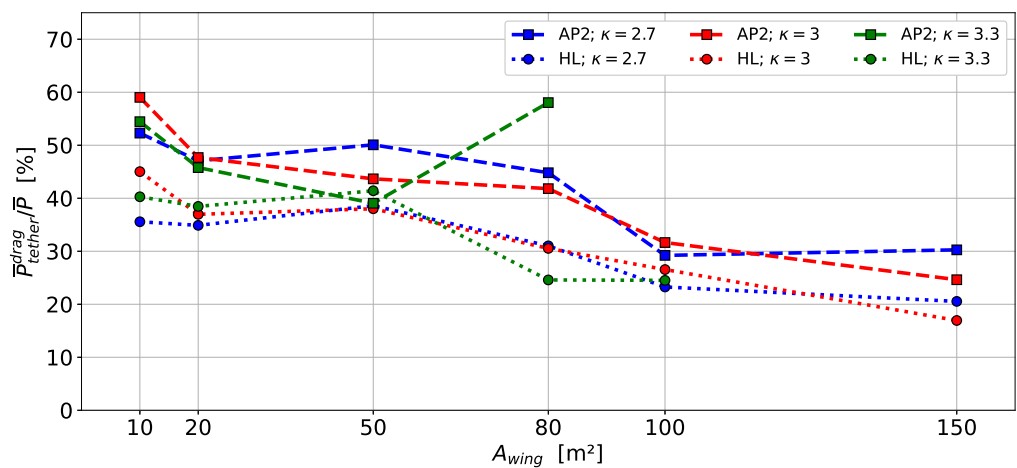

**Figure 16.** Ratio of average cycle power losses due to tether drag $\overline{P}_{\text{tether}}^{\text{drag}}$ to produced power $\overline{P}$ over aircraft size $A$ for both sets of aerodynamic coefficients AP2, HL, all mass scaling exponent of $\kappa = 2.7, 3.0, 3.3$ and wind data at the offshore location.

## 5 Summary and conclusion

This study presented rigid wing AWES scaling trends based on the Ampyx AP2 reference subject to representative onshore
(Pritzwalk in nothern Germany) and offshore (FINO3 research platform in the North Sea) wind conditions. Generator limitations on speed, torque and power were indirectly implemented by setting a fixed tether reeling speed range and diameter of
every design (size and aerodynamic coefficients). This resulted in a constant maximum power and a power curve as a function
of wind speed. We evaluated the impact of wing area and mass scaling as well as nonlinear aerodynamic properties on optimal
trajectories, reaction forces and moments, power generation and AEP, based on the `awebox` power and trajectory optimization model. The `awebox` framework models a 6 DOF aircraft with non-linear aerodynamic coefficients and a multi-node,
rigid tether to which drag and weight are applied. Based on initial conditions, such as elevation angle and tether length, and
constraints, e.g. tether tension and flight envelop limitations, power optimal trajectories are calculated. A representative set of
k-means clustered onshore and offshore wind velocity profiles, derived from the mesoscale WRF model, were used to define
wind boundary conditions.

We analyzed the performance for two sets of nonlinear aerodynamic coefficients, the AP2 reference and a high-lift configuration where AP2 coefficients were modified as if high-lift devices were attached. Wing areas between $A = 10 - 150$ m, with
mass properties scaled according to a geometric scaling law with three different mass scaling exponents $\kappa = 2.7, 3.0, 3.3$, were
implemented into the `awebox` power and trajectory optimization toolbox.

We discussed the impact of mass and system size on typical trajectories and time series data which confirms that instantaneous power can drop to zero during the reel-out phase. This is caused by insufficient lift as the aircraft tries ascent and
maintain tether tension. The minimum wind speed to sustain positive power production during the reel-out phase as well as
tether length and average operating altitude increase with system size and weight. However, operating heights beyond 500 m
are rare and mostly occur as the system depowers above rated wind speed to stay within tether force and flight speed constraints.
Therefore, it could be reasonable to keep the maximum tether length and operating altitude below those values to reduce costs
and permitting burdens. As these constraints become active, the resulting trajectory deforms and diverge from the expected
paths seen for lower wind speeds. This is especially true for high-lift configurations as they reach these limits faster. er losses,

We determined that rated power scales linearly with wing area, assuming that the tether reeling speed constraints are kept
constant and the tether diameter is adjusted appropriately. We chose to size the tether diameter so that rated power is achieved
at about $U_{\text{ref}} = 10 \text{ ms}^{-1}$, independent of size, mass and location. A larger tether diameter would increase rated power and shift
rated speed towards higher wind speeds, which might be beneficial for faster offshore wind conditions, but would impact tether
drag and weight. Higher aerodynamic efficiency increases power production. Average power increased by approximately 30 %
to 80 %, for the sets of aerodynamic coefficients used in this study, depending on wing area.

We estimated AEP and cf based on the power curve analysis and wind speed probability distribution at reference height
between $100 \leq z_{\text{ref}} \leq 400$ m. Offshore AEP were generally higher than onshore, while the power curves are almost identical
even though clustered profiles differ, due to higher wind speeds. Increased aircraft mass lead to a significant reduction in AEP,
as lower wind speeds become infeasible to fly in until finally no feasible solutions, even at higher wind speeds, was found. This

is particularly true for the onshore location and AP2 reference aerodynamics, as these conditions cannot produce sufficient lift force to overcome system weight. Wind farm setups might therefore benefit from the deployment of multiple smaller AWES rather than few large-scale AWES. This could also reduce the overall power loss when phase-shifting the flight trajectories of AWESs within a farm. Determining the ideal, site-specific AWES size needs to be determined subject to realistic mass scaling, the available area and the local wind resource.

Furthermore, we described the tether contribution to total weight and drag relative to aircraft wing size as well as tether-associated power losses. Our results show that even though relative tether power losses decrease with wing size, they still use up a significant portion (20 - 60 %) of the average mechanical AWES power.

Lastly, we investigated the maximum AWES weight-to-lift ratio. Our data showed that total AWES weight, including tether and aircraft, should not exceed 20 % of the produced aerodynamic lift to operate. The limitation of crosswind AWES operations seems to be the upward climb within each loop. During this ascent the aircraft decelerates by approximately 20 %- 25 %, which reduces aerodynamic lift by about 35 % - 45 %, which could be offset by the deployment of additional high-lift devices. As a result, the system cannot produce enough lift to overcome gravity and maintain tether tension, leading to a reduction in tether reeling speed and power up until a complete drop to zero for lower wind speeds. In comparison, conventional WT power scales with the square and mass with the cube of the rotor diameter. Under the same assumptions rigid wing AWES performance scales worse because the aircraft needs to carry the entire increasing system weight (including tether mass), instead of being supported by a tower. Therefore, the optimal AWES size is always defined by the maximum system weight, including tether and aircraft, which the aircraft can support, subject to local wind conditions.

# 6 Future work

It is crucial to investigate the AWES design space subject to realistic wind conditions and operating constraints to further the development of this technology for large-scale deployment. We therefore propose to build upon this study and further investigate the design space using design optimization. A possible approach is to utilize the already existing AWES power and trajectory optimization toolbox `awebox` and implement it into a design optimization framework that varies parameters such as aspect ratio, wing area and wing box dimensions. Adding a cost model would allow to optimize for levelized cost of electricity or AEP. Analyzing the dynamic aircraft wing loads caused by the cyclic nature of crosswind AWES and turbulence could improve AWES durability and further explore AWES design by considering fatigue loads to explore wing concepts to minimize $\kappa$. The sensitivity of the `awebox` model performance to both the number of loops and flight time needs to be verified and compared to other models. Ultimately, AWES must compete with conventional wind. Scaling and moving offshore are logical goals for both technologies. The relative merits of large-scale AWES must be further explored to set design and development targets. This particularly applies to offshore, where they are in direct competition with WTs, as they both operate at lower altitudes, given the generally lower wind speed. This further highlights that the advantage of ground-generation AWES, in particularly offshore, does not lie in higher altitudes, but in reduced material and associated benefits such as easier transportation.

*Author contributions.* Markus Sommerfeld evaluated the data and wrote the manuscript in consultation and under the supervision of Curran Crawford. Martin Dörenkämper set up the numerical offshore simulation, contributed to the meteorological evaluation of the data. Jochem De Schutter co-developed the optimization model and helped writing and reviewed this manuscript.

*Competing interests.* We declare that neither the author nor any co-author can identify any conflict of interest regarding the subject matter or materials discussed in this manuscript.

*Acknowledgements.* The authors thank the BMWi for funding of the "OnKites I" and "OnKites II" projects [grant numbers 0325394 and 0325394A] on the basis of a decision by the German Bundestag and project management Projektträger Jülich. We thank the PICS, NSERC and the DAAD for their funding.

`awebox` has been developed in collaboration with the company Kiteswarms Ltd. The company has also supported the `awebox` project through research funding. The `awebox` project has received funding from the European Union's Horizon 2020 research and innovation 600 program under the Marie Sklodowska-Curie grant agreement No 642682 (AWESCO).

We thank the Carl von Ossietzky University of Oldenburg and the Energy Meteorology research group for providing access to their High Performance Computing cluster *EDDY* and ongoing support.

We further acknowledge Rachel Leuthold (University of Freiburg, SYSCOP) and Thilo Bronnenmeyer (Kiteswarms Ltd.) for their help in writing this article and great technical `awebox` support.

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
