# Peer review of "Scaling effects of fixed-wing ground-generation airborne wind energy systems"

_Wind Energy Science, 2020_

## Short Comment (SC1) · 21 Nov 2020

Thank you so much for this valuable work!! The findings corroborate well with previous works and enhance the scope of knowledge in pumping cycle single wing AWES scaling. You've obviously researched and used the available analysis tools well.

I have a problem with the overriding framing of the work however From the title it looks like the article will be Ground-generation airborne wind energy design space exploration

The title is countered, and a better title is suggested in the opening paragraph This study investigates the scaling effects of single-wing, ground-generation AWES

The paper however also only resolves scaling a single class of groundgen AWES de-

sign (yo-yo or pumping) not the design space.

The article does address the benefits of multi wing systems. Well done. But no attempt is made to describe their operational modes or design space.

By the closing chapters the terms AWES and Crosswind AWES are used as if they can only be describing 1 type of design. There are many potential designs in AWES barely mentioned in the AWES scientific community literature. I can't blame the author for not having been introduced to them. There seems to be a systemic problem of lack of engagement in multi-kite and network scaling effects in AWES scientific academia.

Markus Your work was spot on. You nailed it. This work will be really useful. The historic academic framing of how AWES are designed has shown in your work. I dearly wish to read a genuine Ground-generation airborne wind energy design space exploration.

---

## Referee Comment (RC1) · Anonymous Referee #1 · 9 Dec 2020

GENERAL COMMENTS

Overall it is a good work. I appreciated reading the paper and I found interesting the results. I believe it needs some more physical explanations and interpretations to be a really high quality work. Here are my comments.

SPECIFIC COMMENTS

————- - Table 1

It seems that all your results above rated power are directly influenced by the tether values because the power is the product of $F_{tether}^{max}$ and $\dot{l}_{tether}$. This should be clearly stated in the text and in the conclusions. A design space exploration would investigate the effect of changing these constraints on the power output. Maybe, consider

changing the title (I agree with the other Comment)

————- - Line 182

Have you tried to start from different initial conditions (i.e. different circular loops) to find the global optimum and avoid local optima?

————- - Line 278

It is not clear to me why higher lift coefficient results in higher rated power when the rated power is just the product between reel-out velocity and tether force, which are constrained. Please clarify it.

————- - Line 301

Does the path length include the reel-in part? Please specify it. Is there any physical reason why the flight path remains constant with wind speed? Is this true also after rated?

————- - Line 304

You mention the minimal turning radius without introducing it. Please do that. The minimal turning radius is quite important. Have you a constraint on this? Can you show/ comment how large is the turning radius compared with the wing span? A too small turning radius divided by wing span results in a big difference is tangential velocity between inner and outer part of the wing.

————- - Line 305

For conventional WTs, the evaluation of Cp always requires the evaluation of the axial induction at the wind turbine. In this work, I suppose, the induction is neglected. Please state it here.

Since for AWESs the induction is typically neglected, the concept of swept area loses its typical meaning. Indeed, the reference area is the kite wing area in Loyd equation.

This is why in AWE, the Power harvesting factor (see "Airborne Wind Energy: Basic Concepts and Physical Foundations",Moritz Diehl in "Airborne Wind Energy", Ahrens, Diehl, Schmehl (2013) - pag 18) is defined and not the Cp you define in this paragraph. I believe that introducing this factor in your considerations and plots would make your conclusions more understandable.

Moreover, the power losses associated to the path are typically associated to the cosine of the opening angle (angle defined by the tether length and the turning radius) powered by 3 (see section 4 of "The Influence of Tether Sag on Airborne Wind Energy Generation", Trevisi et al, (2020) for more details). It could be interesting to include this term in your considerations.

———- - Line 390

Why half of the tether drag and the intere tether mass? I do not see the physical reason. Can you elaborate?

———- - Line 405

Did you use include the tether drag in consistent way with literature for the evaluation of the system glide ratio ($\frac{L_{wing}}{D_{total}}$)? For instance see equation 4.8 in "Efficiency of Traction Power Conversion Based on Crosswind Motion",Ivan Argatov and Risto Silvennoinen in "Airborne Wind Energy", Ahrens, Diehl, Schmehl (2013). Using the mentioned equation can be a simple check for your results.

———- - Line 424

Can you explain how you computed the tether drag power loss?

TYPING ERRORS

———- - Line 427

SIM

---

## Referee Comment (RC2) · Anonymous Referee #2 · 29 Dec 2020

**1   General comments**

– Comment 1
In this paper, Sommerfeld et al present a good body of work combining realistic wind data with an optimal control based trajectory optimisation toolbox to explore the effects of multiple wing scale variants.
However, I would advise rephrasing the title as the content is not a "Ground-generation airborne wind energy design space exploration" but rather a comparison of scaling effects based on a reference design that is derived from previous work on the Ampyx AP2 model. The current title promises much more, and as a minimum, a design space exploration should look at more than a single configuration that is just scaled.

[Figure]

– Comment 2

The work presented here is a high level description of the optimisation results for the scaled variants of the AP2 reference model. However, the discussion of the results at times lack depth. The results are simply presented as optimisation results from the model as compared to an actual design/variant exploration which should try and correlate the design variations to the results. The discussions do not distinguish between model and physical/design driven artefacts.

In a few instances, hypotheses for the results obtained are given but they are not fully explored and confirmed. This should be trivial given the capabilities of the model utilised.

In its current form, this manuscript is a macro description of the results of the power optimisation toolbox utilised, and lacks the exploration and discussion of results that would elevate this to a holistic scientific publication.

**2 Specific comments**

– Line 59:
Please clarify what these other studies refer to.

– Line 91:
Have the results between this higher fidelity time resolved model been compared to a simpler steady state model for power prediction? Such a comparison would help motivate the use of such a non-trivial model for a "design space exploration", where it is not immediately visible what effects result from the artefacts of the optimisation algorithm choices, and what are driven directly by the physics of the design.

– Line 132:
What are the actual constraints applied to the tether speeds? Is it $v_{out} = 10m/s$?

– Line 140:

The $C_L$ vs $\alpha$ curves shown depict the typical offset in lift from slats, while in the text flaps are also mentioned. Which one is it?

If the coefficients are just adjusted in a representative manner please mention that, as the change in $C_D$ does not look like the typical change associated with slats. In fact, the $C_D$ looks simply offset as well, which would not be the case with slats.

If this is strictly a theoretical exercise, it would be prudent to ensure that such a combination of resulting $C_L$, $C_D$, and $C_m$ is actually possible from said high lift devices. If this has already been done (or not done), please mention it explicitly. Even simple literature references should suffice for this.

– Lines 165-170:

The actual constraints utilised for the results discussed is vital and missing. Are all the system constraints described in Section 3.6 in Table 1.? While implementation specific constraints are also helpful, as a bare minimum, system constrains should be listed clearly. Interpreting optimisation results without the specifics of the constraints can be quite futile.

– Line 202:

It is not clear why the results should not be consistent if all the studies referenced to are utilising the same model. Describe the reasoning behind expecting different results if the path constraints are the same?

– Lines 203-205:

Please add a plot with the tether forces and speeds over the cycle, to show how the expected trajectory is changed in order to satisfy the constraints? This would make things much clearer.

– Line 215:

Why is total cycle time constant across all variants explored? This seems more like an artefact of how the problem is setup/constraints placed on it, rather than something

stemming from the physics of the problem. Please clarify the reason for this constant cycle time.

– Line 216:
Kindly cite the previous investigations - again if they all utilise the same model, it should be highlighted that this could be a model based artefact.
While from these results it would seem that the cycle time has no impact on the power produced, the trickle down effect to more detail design drivers such as structural loads (as touched upon in 4.3) are highly impacted by loading frequency which would be a function of the cycle time. Hence it would be good to know if this result is something physical or just model based.

– Lines 219-222:
Some more clarity on the aerodynamic forces periodicity would be helpful here. How does it correlate to only wind speed variations for the same scale of aircraft. Similarly, how does it correlate to aircraft scale for a given wind speed? What can be said about the influence of cycle time on the periodicity? These kind of discussions are expected from such work rather than just a direct description of the results.

– Line 230:
It would be good to clarify that the power factor ratio considered ($C_L^3/C_D^2$) doesn't contain the tether drag. Instead, it might be prudent to use the total system's power factor that includes the tether drag when interpreting these results.

– Figure 4:
It would seem that the model always tries to maintain the same number of loops (5?). Please clarify why this might be?

This figure would be much more useful if the variant wing design shown in Appendix A1 was included in the comparison.
Results from A1 are for a different (onshore) wind profile, that makes it meaningless to try and derive any conclusions b/w the two wing variants (reference AP2 and AP2
HL). In the current form, the figure lacks clarity. Instead, a single figure showing the trajectories for multiple variants followed by a more detailed figure investigating the variations along the full trajectory would be much more interesting.

– Section 4.3:

-Line 260:

Are the forces from manoeuvre loads due to the loops also neglected in this section? Was an initial estimate made before neglecting these effects? Is this true for all scales of wings considered in this study?

Given the nature of the study (path optimisation and wing scaling), can anything be said about the significance of ignoring the loads arising directly from the path? Overall, the motivation for including this section is lacking. The results presented here are straight forward and well understood - forces and moments scale with the wing span and aerodynamic lift coefficient. Please consider dropping this section as it does not contribute any new insights.

– Line 283:

How exactly is this $\bar{P}_{rated}$ determined? Please clarify.

Is it constrained by the model? The tether forces and speeds are constrained from my understanding.

– Line 291:

Do mention effects of mass and resulting cut-in wind speed on the AEP here as well.

– Line 298:

Please motivate the depowering hypothesis for the power variations. Taking the case of $A_{wing}50$, why does the system depower already around the rated wind speed? $\bar{U}_{ref} = 10$? An additional plot of the constraint violations at such points would be very insightful.

– Line 301:

The discussion on path length is quite lacking. From figure 7, there seems to be quite some variation that is not explained.

For example, what could be the reason for increase in $I_{path}$ for the $A_{wing}50$ variant at wind speeds above 25? Is it the result of a depowering manoeuvre? Please explain these results better.

– Line 304:
Another reason why system constraints utilised in the model should be explicitly mentioned in one place. How is the minimal turning radius defined? How does it scale with the wings aspect ratio? Is it constrained?

– Figure 7:
Please add the configuration (HL?) utilised to the legend.

Rated power plots seem to have quite some anomalies - do mention if/how the data is interpolated. For example $A_{wing}150$ seems to have data points that are not on the curve.

– Line 305:
The motivation for defining this $C_p^{AWES}$ is missing. The swept area for AWES doesn't compare well to the swept area of traditional wind turbines.

– Line 312:
Please motivate such hypothesis with more data, it should be trivial with the model utilised here to investigate the ratio of lift force to tether drag for the design variants considered.

A figure plotting the aerodynamic forces, inertia forces and tether drag for a few wing variants considered in the study would greatly enhance the rather brusque discussion of these results.

– Line 336:
Please consider adding another data point for either configuration with a different mass scaling factor in addition to the AEP variation for the reference vs HL configuration. Figure 10 does explore this aspect to some extent, but it could be combined here for a

better discussion.

– Figure 8:

What is the reason for the HL configuration not to follow the linear increase in rated power until $U_{ref} = 10$ as the AP2 configuration? There seems to be quite some perturbations at lower wind conditions?

– Line 344:

Clarify what aspect of Figure 7 shows the infeasible solutions for lower wind speeds.

– Line 390:

Why is half the tether drag attributed to the ground station? How is the ground station affected by the tether drag?

– Line 392:

Doesn't the 6DOF model utilised here compensate for the reduction in apparent speed by increasing angle of attack to maintain the same aerodynamic force?

– Line 405:

Again, why is half the drag assigned to the ground station? The total system drag is of importance here.

– Line 411:

Please add additional data to back this hypothesis. Would this change if the cycle time changes? This seems again an artefact of the model implementation, and not a physical effect. If this is not the case, please motivate.

– Line 440:

This should include the system weight (including tether mass).

– Lines 445,455:

Some ambiguity in the nomenclature - high lift airfoil != high lift devices (as was described previously).

– Lines 460:
Would this be different if tether speeds are varied instead of being fixed? Is this fixed tether speed assumption probably the reason why the model has fixed cycle times for all the design variants considered?

**3   Technical corrections**

– Line 11:
Missing space

– Line 427:
sim=> ∼

---

## Referee Comment (RC3) · Anonymous Referee #3 · 6 Jan 2021

General =======

The paper addresses effects on the scaling-up of AWES. This is a timely and important topic in the field.

WES Criteria ============

1. Does the paper address relevant scientific questions within the scope of WES?

Yes.

2. Does the paper present novel concepts, ideas, tools, or data?

Yes.

3. Is the paper of broad international interest?

Yes.

4. Are clear objectives and/or hypotheses put forward?

This can clearly be improved. In the introduction, the authors can better highlight what the main hypothesis is (one paragraph) and what the contributions of this paper are (one paragraph).

5. Are the scientific methods valid and clear outlined to be reproduced?

In principle, reproduction is complex but should be possible.

6. Are analyses and assumptions valid?

Yes (see questions/discussions below).

7. Are the presented results sufficient to support the interpretations and associated discussion?

Yes (see questions/discussions below).

8. Is the discussion relevant and backed up?

Yes (see questions/discussions below).

9. Are accurate conclusions reached based on the presented results and discussion?

Yes (see questions/discussions below).

10. Do the authors give proper credit to related and relevant work and clearly indicate their own original contribution?

Yes.

11. Does the title clearly reflect the contents of the paper and is it informative?

The authors may consider changing the title e.g. to "Scaling effects of rigid kite ground-

generation airborne wind energy".

12. Does the abstract provide a concise and complete summary, including quantitative results?

Yes (with the same limitations as discussed above and below).

13. Is the overall presentation well structured?

Yes.

14. Is the paper written concisely and to the point?

Yes.

15. Is the language fluent, precise, and grammatically correct?

Yes (some improvements given below).

16. Are the figures and tables useful and all necessary?

Yes.

17. Are mathematical formulae, symbols, abbreviations, and units correctly defined and used according to the author guidelines?

Yes.

18. Should any parts of the paper (text, formulae, figures, tables) be clarified, reduced, combined, or eliminated?

No.

19. Are the number and quality of references appropriate?

Yes.

20. Is the amount and quality of supplementary material appropriate and of added value?

Yes.

Questions/Discussions =====================

Figure 3: Do these curves originate from CFDs or wind tunnel tests? From the text it sounds like the characteristic is constructed/guessed "by hand" (also indicated by the unrealistically high negative lift coefficients). – It is important to have solid aerodynamic characteristics as the sensitivity of those on the power/energy/economics is high. If the curves in Fig. 3 are polynomial simplifications based on CFDs/wind tunnel data, please plot that original data also into the graphs.

Line 230: Angle of attack and thus lift coefficient seems to increase with the wind speed. This is unexpected to me. I'd rather expect the either the exact opposite to limit loads at high wind, or that the angle of attack remains mainly constant for all wind speeds during reel-out. Can you explain why the lift is changed so much? Can you also plot the apparent airspeed of the aircraft?

Line 259: Weight is neglected -> please clarify; the text before explains how it is accounted for

Table 1: Where does the value for d_tether originate from? Should it not be left to the optimizer to find the optimal value (given the constraint that a lower tether diameter limits F_tether^max)? Also, it could have been left to the optimizer to find the optimal rated wind speed.

Line 347: This is likely caused by outliers, or wind velocity profile specific local minima -> In the paper, this is often said. How much trust can a reader give to the results, not knowing if it is a local or global minimum? (Would it be worth using simulation models and algorithms which can find the global optimum like swarm-optimization algorithms?)

Line 452: However, operating heights beyond 500 m are rare and mostly occur as the system de-powers above rated wind speed to stay within tether force and flight speed constraints. -> Is it possible and meaningful to keep the maximum tether length and
operating altitude below those values to reduce costs and permitting burdens?

Minors ======

Line 12: we estimate a minimum average cycle-average lift to weight ratio -> we estimate a minimum cycle-average lift to weight ratio (?)

Line 21: This study focuses on the two-phase, ground-generation concept -> You might consider a full stop there and delete everything until end of line 25. No need to list (apparent) drawbacks of drag power.

Line 35: re-power decommissioned offshore wind farms or deploy floating platforms -> is this correct? source?

Line 66: for for

Line 135: If the coefficients are meant not for the 2D airfoil, you may consider using capital letter C instead of lower case c. Note that $C_L/C_D$ and $C_L^3/C_D^2$ has only a meaning for untethered flight or if $C_D$ is for aircraft+tether.

Line 165: to reduce the mechanical wing load -> to limit ... (question: why not imposing a constraint on the wing loading directly instead?)

Line 168: but implemented as tether speed, acceleration constraints -> but implemented as tether speed and acceleration constraints (?)

Line 187: Results -> Results and Discussion (?)

Line 200: However, ... -> please double-check language

Line 202: Consider replacing the phrase "It is striking"

Line 204: with in -> within (?)

Line 222: it's -> its (double-check entire paper for this)

Line 237: in higher drag losses and -> in higher drag losses or

[Figure]

Line 237: the cosine loss due elevation angle is not caused by gravity (remove "gravity-caused")

Line 265: what is meant by "maximum cycle-average loads"? -> maximum load during a cycle?

Line 292: "only cut-in wind speed" seems lost

Line 335: lead result

Line 359: Determining ... determined

Line 370: power only scales with the wing area (F_lift âĹij bˆ2) -> note that the tether diameter and thus tether drag scale slower which is why power should scale faster than with bˆ2.

Line 427: sim -> \sim

Line 457: a elliptical lift distribution -> an elliptical lift distribution

Line 471: do can not produce

Line 495: offshore AWES are not particularly beneficial relative to conventional wind, given the generally lower sheer offshore -> Note that this is just another confirmation of the fact, that AWES advantage (at least for this concept), in particularly offshore, lies not in higher altitudes but reduced building material and associated benefits (transport etc.).

---

## Author Comment (AC1) · 31 Dec 2021

article [utf8]inputenc [utf8]inputenc lmodern xcolor [normalem]ulem geometry float lipsum multirow hyperref tikz

**Response to referee 1 wes-2020-123**

**Markus Sommerfeld**

**December 31, 2021**

**1 Author response**

Dear reviewer 1,

Thank you very much for your helpful comments to our manuscript, "Ground-generation airborne wind energy design space exploration", wes-2020-123. I am very sorry that I could not resubmit the revision sooner. My full-time work and family, together with a relocation to Japan, required my attention and time.

The manuscript underwent major revision. Several figures and sections have been replaced and new ones have been added.

Sincerely, Markus Sommerfeld

**2 General Comments**

Overall it is a good work. I appreciated reading the paper and I found interesting the results. I believe it needs some more physical explanations and interpretations to be a really high quality work. Here are my comments.

**3 Specific comments**

Table 1 It seems that all your results above rated power are directly influenced by the tether values because the power is the product of $F_{tether}^{max}$ and $\dot{l}_{tether}$. This should be clearly stated in the text and in the conclusions. A design space exploration would investigate the effect of changing these constraints on the power output. Maybe, consider changing the title (I agree with the other Comment)

– Changed title

– Added clarification in sub-section Tether model.

– Varying tether diameter and thereby maximum tether force would vary rated power and rated wind speed. An interesting analysis would be to determine the optimal tether size for a given location to maximize AEP.

– At higher wind speeds, a higher max. reel-out speed could would increase instantaneous power. At low wind speeds ideal reel-out speed (according to Loyd with cosine losses) is within the current constraint. Choosing the right ground station design (drum diameter, motor, winch etc.) is an optimization problem beyond the scope of this paper.

Line 182 Have you tried to start from different initial conditions (i.e. different circular loops) to find the global optimum and avoid local optima?

– To determine the global optimum with certainty, using the current optimizer, is not possible due to the complexity of the problem. Varying the initial conditions to determine the global optimum would also drastically increase the computational cost as it would have to be adjusted for each of the 6 designs, 3 weights, 30 wind conditions, two locations and two sets of aerodynamic coefficients.

– The initial path is generated via several homotopy strategy which transform an initial circular path to a periodic path similar to a helix. The radius of the initial circle is determined from the tether length and estimated flight speed. We compared several initial tether lengths depending on system mass, size and wind speed to achieve convergence. We found that initial tether lengths needs to increase with system size and wind speed to converge to a feasible solution.

– Clarified in text

Line 278 It is not clear to me why higher lift coefficient results in higher rated power when the rated power is just the product between reel-out velocity and tether force, which are constrained. Please clarify it.

– Aerodynamic coefficients indirectly affect rated power because the tether diameter, and therefore tether force, is changed such that rated wind speed is kept constant. Tether reel-in and reel-out are kept constant for all designs.

– Clarified in text

Line 301 Does the path length include the reel-in part? Please specify it. Is there any physical reason why the flight path remains constant with wind speed? Is this true also after rated?

– This section has been re-written and $c_p$ has been replaced with the power harvesting factor https://link.springer.com/chapter/10.1007%2F978-3-642-39965-7_1

Line 304 You mention the minimal turning radius without introducing it. Please do that. The minimal turning radius is quite important. Have you a constraint on this? Can you show/ comment how large is the turning radius compared with the wing span? A too small turning radius divided by wing span results in a big difference is tangential velocity between inner and outer part of the wing

– Paragraph removed.

– The minimal turning radius is the result of optimization and not a fixed constraint. Larger systems with higher inertia have a larger turning radius.

– Determining the turning radius is difficult as trajectories are so diverse and deteriorate at higher wind speeds.

Line 305 For conventional WTs, the evaluation of Cp always requires the evaluation of the axial induction at the wind turbine. In this work, I suppose, the induction is neglected. Please state it here. Since for AWESs the induction is typically neglected, the concept of swept area loses its typical meaning. Indeed, the reference area is the kite wing area in Loyd equation. This is why in AWE, the Power harvesting factor (see "Airborne Wind Energy: Basic Concepts and Physical Foundations",Moritz Diehl in "Airborne Wind Energy", Ahrens, Diehl, Schmehl (2013) - page 18) is defined and not the Cp you define in this paragraph. I believe that introducing this factor in your considerations and plots would make your conclusions more understandable. Moreover, the power losses associated to the path are typically associated to the cosine of the opening angle (angle defined by the tether length and the turning radius) powered by 3 (see section 4 of "The Influence of Tether Sag on Airborne Wind Energy Generation", Trevisi et al, (2020) for more details). It could be interesting to include this term in your considerations

– Removed the calculation of $c_p$ and replaced with the power harvesting factor from Diehl et al.

– re-wrote this section

Line 390 Why half of the tether drag and the entire tether mass? I do not see the physical reason. Can you elaborate?

– Mentioning half the tether drag here was supposed to reference the previous description in Sub-section "Tether model". This Sub-section explains that

half of the tether drag at every element is equally divided between the two endpoints and finally transferred to either the aircraft or ground station.

– Clarified in the text.

– The entire tether mass is assigned to the aircraft node, because it is the only thing supporting it and keeping it in the air. As the tether can not support compressive forces, the ground station can not support it.

Line 405 Did you use include the tether drag in consistent way with literature for the evaluation of the system glide ratio ($\frac{L_{wing}}{D_{total}}$)? For instance see equation 4.8 in "Efficiency of Traction Power Conversion Based on Crosswind Motion",Ivan Argatov and Risto Silvennoinen in "Airborne Wind Energy", Ahrens, Diehl, Schmehl (2013). Using the mentioned equation can be a simple check for your results. ?

– The shown data is the result of the optimization. The tether model is described in section 3.5 'Tether model'

– I did not include a comparison to the simplified tether drag estimate used in the QQS model in section 3.2, equation 3.

– I believe including this requires the generation of another figure as including these data would overcrowd the plot?

– Do you believe the inclusion is necessary for the understanding of these results?

Line 242 Can you explain how you computed the tether drag power loss?

– Added a clarification in Sub-section Power losses

– A portion of the tether drag is attributed to the kite node ($F_{tether-drag}^{kite}$), which is as we know an underestimation of the actual tether drag. The kite moves at a speed of $v_{kite}$. Then the power loss is defined as $F_{tether-drag}^{kite} \cdot v_{kite}$.

Line 427  TYPING ERRORS SIM

    – corrected

Interactive
comment

---

## Author Comment (AC2) · 31 Dec 2021

I am sorry, the previous message was for RC1.

---

## Author Comment (AC4) · 31 Dec 2021

article [utf8]inputenc [utf8]inputenc lmodern xcolor [normalem]ulem geometry float lipsum multirow hyperref tikz
**Response to referee 2 wes-2020-123**

Markus Sommerfeld

December 31, 2021

**1 Author response**

Dear referee 2,

Thank you very much for your helpful comments to our manuscript, "Ground-generation airborne wind energy design space exploration", wes-2020-123. I am very sorry that I could not resubmit the revision sooner. My full-time work and family, together with a relocation to Japan, required my attention and time.

The manuscript underwent major revision. Several figures and sections have been replaced and new ones have been added.

Sincerely, Markus Sommerfeld

**2 General comments**

Comment 1 In this paper, Sommerfeld et al present a good body of work combining realistic wind data with an optimal control based trajectory optimisation toolbox to explore the effects of multiple wing scale variants. However, I would advise rephrasing

the title as the content is not a "Ground-generation airborne wind energy design space exploration" but rather a comparison of scaling effects based on a reference design that is derived from previous work on the Ampyx AP2 model. The current title promises much more, and as a minimum, a design space exploration should look at more than a single configuration that is just scaled.

– Changed title

Comment 2 The work presented here is a high level description of the optimisation results for the scaled variants of the AP2 reference model. However, the discussion of the results at times lack depth. The results are simply presented as optimisation results from the model as compared to an actual design/variant exploration which should try and correlate the design variations to the results. The discussions do not distinguish between model and physical/design driven artefacts. In a few instances, hypotheses for the results obtained are given but they are not fully explored and confirmed. This should be trivial given the capabilities of the model utilised.

In its current form, this manuscript is a macro description of the results of the power optimisation toolbox utilised, and lacks the exploration and discussion of results that would elevate this to a holistic scientific publication.

– Tried to address and added additional figures.

**3 Specific comments**

Line 59 Please clarify what these other studies refer to.

– Added citations

Line 91 Have the results between this higher fidelity time resolved model been compared to a simpler steady state model for power prediction?

– added a reference sub-section "Quasi-steady state reference model" and a comparison to optimized power in the results section.

Line 132 What are the actual constraints applied to the tether speeds? Is it $v_{out} = 10m/s$?

– Most of the constraint are mentioned Section 3.6 Constraints and Table 1, $v_{out} = -15 : 10 \ m/s$

Line 140 The $CL$ vs $\alpha$ curves shown depict the typical offset in lift from slats, while in the text flaps are also mentioned. Which one is it? If the coefficients are just adjusted in a representative manner please mention that, as the change in $CD$ does not look like the typical change associated with slats. In fact,the $CD$ looks simply offset as well, which would not be the case with slats. If this is strictly a theoretical exercise, it would be prudent to ensure that such a combination of resulting $CL$, $CD$, and $Cm$ is actually possible from said high lift devices. If this has already been done (or not done), please mention it explicitly. Even simple literature references should suffice for this.

– added references
– yes, $CD$ is just offset by adding to $CD_0$. $CL_0$ is offset as well and stall is delayed (see non-extending flaps Figure 8.1 and leading edge slot 8.3: https://www.fzt.haw-hamburg.de/pers/Scholz/HOOU/AircraftDesign_8_HighLift.pdf).
– I struggled finding a better refernce for the effect of HL on drag, can you please recommend one? In the end, the point of this investigation is to identify the effect of improved aerodynamic coefficients on AWES performance. Therefore, I could also remove mentioning high lift devices.

- The text mentions: 'Lift and drag at zero angle of attack are increased, stall is delayed, and pitch moment decreased.'

Line 165-170 The actual constraints utilised for the results discussed is vital and missing. Are all the system constraints described in Section 3.6 in Table 1.? While implementation specific constraints are also helpful, as a bare minimum, system constrains should be listed clearly. Interpreting optimisation results without the specifics of the constraints can be quite futile.

- Added reference to Table 1, which summarizes the majority of the implemented constraints, earlier in the text
- Inequality constraints: angle of attack and side slip angle or tether speed, acceleration and operating altitude.
- Equality constraints: Tether diameter, density and aircraft size and weight
- Added a citation to the relevant awebox website and publications for further information
- Which other constraints do you see as necessary to be specifically mentioned / listed to improve understanding?

Line 202 It is not clear why the results should not be consistent if all the studies referenced to are utilising the same model. Describe the reasoning behind expecting different results if the path constraints are the same?

- Rewritten sentence. This sentence just signifies that the trajectories are not irregular, follow expected paths. I am not expecting different results. The fact that they are consistent with previous results just emphasizes that these are not exceptional or hand-picked examples.

Lines 203-205 Please add a plot with the tether forces and speeds over the cycle, to show how the expected trajectory is changed in order to satisfy the constraints? This would make things much clearer.

– Broke up the figure into 2 figures and added additional sub-figures to compare the changes in trajectory and caused by changes in wind speed and aerodynamic coefficients.

Line 215 Why is total cycle time constant across all variants explored? This seems more like an artefact of how the problem is setup/constraints placed on it, rather than something stemming from the physics of the problem. Please clarify the reason for this constant cycle time.

– Very minor variation between aerodynamic coefficient and wind speed were observed in the previous awebox version. The updated awebox now result higher variation in cycle times depending on system configuration. The plots have been updated and commented in the text.

Line 216 Kindly cite the previous investigations - again if they all utilise the same model, it should be highlighted that this could be a model based artefact. While from these results it would seem that the cycle time has no impact on the power produced, the trickle down effect to more detail design drivers such as structural loads (as touched upon in 4.3) are highly impacted by loading frequency which would be a function of the cycle time. Hence it would be good to know if this result is something physical or just model based.

– Clarified in text: These were previous comparisons swiping over a number of loops.

– The investigation of load cycles is mentioned in the future work section

es 219-222 Some more clarity on the aerodynamic forces periodicity would be helpful here. How does it correlate to only wind speed variations for the same scale of aircraft. Similarly, how does it correlate to aircraft scale for a given wind speed? What can be said about the influence of cycle time on the periodicity? These kind of

discussions are expected from such work rather than just a direct description of the results.

– Added additional figure describing the drops in tether tension (tension troughs)

Line 230 It would be good to clarify that the power factor ratio considered ($CL^3/CD^2$) does not contain the tether drag. Instead, it might be prudent to use the total system's power factor that includes the tether drag when interpreting these results.

– Clarified in text
– implement tether drag power estimate & tether drag estimate in QSS section

Figure 4 It would seem that the model always tries to maintain the same number of loops (5?). Please clarify why this might be? This figure would be much more useful if the variant wing design shown in Appendix A1 was included in the comparison. Results from A1 are for a different (onshore) wind profile, that makes it meaningless to try and derive any conclusions b/w the two wing variants (reference AP2 and HL) In the current form, the figure lacks clarity. Instead, a single figure showing the trajectories for multiple variants followed by a more detailed figure investigating the variations along the full trajectory would be much more interesting.

– Clarified in text
– Number of loops is fixed = constraint
– Split figure into 2 and added additional reference for to better contextualize the results.

Line 260 Are the forces from manoeuvre loads due to the loops also neglected in this section? Was an initial estimate made before neglecting these effects? Is this true for all scales of wings considered in this study? Given the nature of the study

(path optimisation and wing scaling), can anything be said about the signifiň- cance of ignoring the loads arising directly from the path? Overall, the motivation for including this section is lacking. The results presented here are straight for- ward and well understood - forces and moments scale with the wing span and aerodynamic lift coefifiň cient. Please consider dropping this section as it does not contribute any new insights.

– Removed this section

Line 283 How exactly is $P_{rated}$ determined? Please clarify. Is it constrained by the model? The tether forces and speeds are constrained from my understanding.

– Clarified in text
– Rated power here is defined as the maximum, almost constant cycle aver- age power, which is constrained by the limits on instantaneous tether force and speed. From this rated wind speed can be inferred.

Line 291 Do mention effects of mass and resulting cut-in wind speed on the AEP here as well.

– Section rewritten and figure replaced

Line 298 Please motivate the depowering hypothesis for the power variations. Taking the case of $A_{wing} = 50$, why does the system depower already around the rated wind speed? $U_{ref} = 10$? An additional plot of the constraint violations at such points would be very insightful.

– Section rewritten and figure replaced
– Variation in power are the result of wind speed profile shape and the AWES trajectory. Formulation was not clear. The AWES needs to depower to stay within tension constraints (adjust angle of attack) which result in constant rated power.

Line 301 The discussion on path length is quite lacking. From figure 7, there seems to
be quite some variation that is not explained. For example, what could be the
reason for increase in $L_{path}$ for the $A_{wing} = 50$ variant at wind speeds above 25?
Is it the result of a depowering manoeuvre? Please explain these results better.

– Section removed and replaced with power harvest factor

Line 304 Another reason why system constraints utilised in the model should be explicitly
mentioned in one place. How is the minimal turning radius defined? How does
it scale with the wings aspect ratio? Is it constrained?

– Section removed and replaced with harvesting factor

Figure 7 Please add the configuration (HL?) utilised to the legend. Rated power plots
seem to have quite some anomalies - do mention if/how the data is interpolated.
For example $A_{wing} = 150$ seems to have data points that are not on the curve.

– Figure removed and replaced with harvesting factor

Line 305 The motivation for defining this $C_p^{AWES}$ is missing. The swept area for AWES
doesn't compare well to the swept area of traditional wind turbines.

– Section removed and replaced with harvesting factor

Line 312 Please motivate such hypothesis with more data, it should be trivial with the
model utilised here to investigate the ratio of lift force to tether drag for the design
variants considered.
A figure plotting the aerodynamic forces, inertia forces and tether drag for a few
wing variants considered in the study would greatly enhance the rather brusque
discussion of these results.

– Section removed and replaced with harvesting factor

Line 336 Please consider adding another data point for either configuration with a different mass scaling factor in addition to the AEP variation for the reference vs HL configuration. Figure 10 does explore this aspect to some extent, but it could be combined here for a better discussion.

– Added another figure with quasi steady-state model for reference

Figure 8 What is the reason for the HL configuration not to follow the linear increase in rated power until $U_{ref} = 10$ as the AP2 configuration? There seems to be quite some perturbations at lower wind conditions?

– Clarified in text
– This is the result of the optimization for various wind profile shapes → different local optima

Line 344 Clarify what aspect of Figure 7 shows the infeasible solutions for lower wind speeds.

– Figure removed
– the missing data represents infeasible solutions

Line 390 Why is half the tether drag attributed to the ground station? How is the ground station affected by the tether drag?

– Clarified in text
– As descried in Sub-section 3.5, the total tether drag is divided up evenly between the top and bottom node at every tether segment, resulting in half the tether drag being attributed to the aircraft and the other half to the ground station.

Line 392 Doesn't the 6DOF model utilised here compensate for the reduction in apparent speed by increasing angle of attack to maintain the same aerodynamic force?

– Clarified in text. Yes, but this is not enough to maintain sufficient lift and tether tension. Compare time series figure 6 and figure 7.

Line 405 Again, why is half the drag assigned to the ground station? The total system drag is of importance here.

– clarified in text: As descried in Sub-section 3.5, the total tether drag is divided up evenly between the top and bottom node at every tether segment, resulting in half the tether drag being attributed to the aircraft and the other half to the ground station.

– The total system drag is compensated by both the aircraft and the ground station, while the system weight is compensated only by the aircraft, because the tether can not support compression. For simplification, half of the tether drag is assigned to both the aircraft and ground station. In reality this will be distributed differently.

Line 411 Please add additional data to back this hypothesis. Would this change if the cycle time changes? This seems again an artefact of the model implementation, and not a physical effect. If this is not the case, please motivate.

– Added a small sentence referencing tension trough analysis.

– What kind of data could back this up? Optimization runs at lower wind speeds did not converge and can not be used for this analysis. It might be possible that AWES with a lower load factor can operate, but power production would be extremely low.

– It is difficult to derive general weight limits because lift force varies so much along the trajectory which is why I used average lift force.

Line 440 This should include the system weight (including tether mass).

– Clarified in text

Line 445,455 Some ambiguity in the nomenclature - high lift airfoil != high lift devices (as was described previously).

- – That is correct. What is meant is that the high lift configuration or high lift airfoil is derived from adjusting the AP2 coefficients as if high lift devices were used.
- – clarified in text

Line 460 Would this be different if tether speeds are varied instead of being fixed? Is this fixed tether speed assumption probably the reason why the model has fixed cycle times for all the design variants considered?

- – Yes, without this constraint the results would look very different. With unconstrained tether speed the produced power would continue to increase with wind speed.
- – This constraint was applied to reflect ground station motor and generator limitations.
- – The design of the ground station needs to strike a compromise between maximum power, AEP or cf and cost. In the end it is another location and wind resource specific optimization.
- – Reran optimization with an updated awebox version. Now cycle times vary more with wind speed and design.

**4 Technical corrections**

Line 11 Missing space

- – implemented

Line 427  sim=>

    – implemented

---

## Author Comment (AC5) · 31 Dec 2021

article [utf8]inputenc [utf8]inputenc lmodern xcolor [normalem]ulem geometry float lipsum multirow hyperref tikz

**Response to referee 3 wes-2020-123**

Markus Sommerfeld

December 31, 2021

**1   Author response**

Dear reviewer 3,

Thank you very much for your helpful comments to our manuscript, "Ground-generation airborne wind energy design space exploration", wes-2020-123. I am very sorry that I could not resubmit the revision sooner. My full-time work and family, together with a relocation to Japan, required my attention and time.

The manuscript underwent major revision. Several figures and sections have been replaced and new ones have been added.

Sincerely, Markus Sommerfeld

**2   General Comments**

Are clear objectives and/or hypotheses put forward? This can clearly be improved. In the introduction, the authors can better highlight what the main hypothesis is (one paragraph) and what the contributions of this paper are (one paragraph).

• Added 2 paragraphs outlining the hypothesis and main contributions.

Does the title clearly reflect the contents of the paper and is it informative? The authors may consider changing the title e.g. to "Scaling effects of rigid kite ground generation airborne wind energy".

• Changed the title

**3  Specific comments**

Figure 3 Do these curves originate from CFDs or wind tunnel tests? From the text it sounds like the characteristic is constructed/guessed "by hand" (also indicated by the unrealistically high negative lift coefficients). – It is important to have solid aerodynamic characteristics as the sensitivity of those on the power/energy/economics is high. If the curves in Fig. 3 are polynomial simplifications based on CFDs/wind tunnel data, please plot that original data also into the graphs.

– Clarified in text: identified in AVL CFD analyses by Ampyx Power and during untethered test flights.

Line 230 Angle of attack and thus lift coefficient seems to increase with the wind speed. This is unexpected to me. I'd rather expect the either the exact opposite to limit loads at high wind, or that the angle of attack remains mainly constant for all wind speeds during reel-out. Can you explain why the lift is changed so much? Can you also plot the apparent airspeed of the aircraft?

– Replaced the figure with an updated version

– Reran the optimization and rewrote the plotting code. The results now make
more sense, with angle of attack decreasing with wind speed to offset the
increased apparent wind speed to stay within the tether force constraint.

Line 259 Weight is neglected → please clarify; the text before explains how it is accounted
for

– This section has been removed in response to a request by a different re-
viewer.

Table 1 Where does the value for $d_{tether}$ originate from? Should it not be left to the opti-
mizer to find the optimal value (given the constraint that a lower tether diameter
limits $F_{tether}^{max}$)? Also, it could have been left to the optimizer to find the optimal
rated wind speed.

– The optimizer is capable of finding the optimal tether diameter. However, not
constraining the tether diameter will result in varying tether diameters and
therefore maximum power with wind speed.

– Instead, we chose to investigate fixed AWES designs at various wind
speeds, mimicking their real-world implementation.

– Yes, implementing an additional optimizer wrapped around the awebox is
possible and mentioned in the future works section of the paper. One could,
for example, optimize AWES for AEP or LCOE which would result in an
optimal wing span and tether diameter. Another approach could be to im-
plement the awebox into a design optimization framework to optimize air-
craft design parameters. However, this was not part of this investigation and
would require considerable more work accompanied by drastically increased
computational cost.

Line 347 This is likely caused by outliers, or wind velocity profile specific local minima →
In the paper, this is often said. How much trust can a reader give to the results,

not knowing if it is a local or global minimum? (Would it be worth using simulation models and algorithms which can find the global optimum like swarm-optimization algorithms?)

- I understand the sentiment and the agree that a lot more can be done. The utilized algorithm only guarantees local minima of a highly complex problem, which should be transparently communicated.
- Determining the global optimum would also drastically increase the computational cost as it would have to be adjusted for each of the 6 designs, 3 weights, 30 wind conditions, two locations and two sets of aerodynamic coefficients. Even more computational cost if aircraft design and tether diameter would be varied to otimize AEP or LCOE.
- Therefore, a decision was made to limit the scope of this investigation on a large, but manageable amount of designs and wind conditions.
- With respect to the AEP of $A_{aircraft} = 150m^2$:
  * AEP tries to represent the power as well as the the annual wind conditions which are (somewhat) arbitrarily chosen from 10 clusters. The particular shape of that wind profile together with the complex model and optimization problem affect the total AEP.
  * Figure **??** and **??**, which are also shown in the appendix of the paper and referenced in the text, show the particular power curves (top) that were used to calculate AEP.
  * Particularly, for $A_{wing} = 150m^2$ you can see how local minima, mall decreases in power at frequent wind conditions, affect the AEP calculation.
  * Filtering out these power dips would likely solve this particular issue, but we chose to keep them in to not distort the actual results.

Line 452  However, operating heights beyond 500 m are rare and mostly occur as the system de-powers above rated wind speed to stay within tether force and flight speed

constraints. → Is it possible and meaningful to keep the maximum tether length and operating altitude below those values to reduce costs and permitting burdens?

- – Yes, this is a good point and it makes sense. I included it in the conclusion

**4 Minors**

Line 12 we estimate a minimum average cycle-average lift to weight ratio → we estimate a minimum cycle-average lift to weight ratio (?)

- – implemented and rewritten

Line 21 This study focuses on the two-phase, ground-generation concept → You might consider a full stop there and delete everything until end of line 25. No need to list (apparent) drawbacks of drag power.

- – Sentences removed

Line 35 Re-power decommissioned offshore wind farms or deploy floating platforms → is this correct? source?

- – Added reference

Line 66 for for

- – implemented

Line 135 If the coefficients are meant not for the 2D airfoil, you may consider using capital letter C instead of lower case c. Note that $C_L/C_D$ and $C_L^3/C_D^2$ has only a meaning for untethered flight or if $C_D$ is for aircraft+tether
– Replaced with capital C to clarify that these are aerodynamic coefficients for the entire aircraft.

Line 165 to reduce the mechanical wing load → to limit ... (question: why not imposing a constraint on the wing loading directly instead?)

– removed sentence and flight speed constraint, because redundant with tether force constraint

Line 168 but implemented as tether speed, acceleration constraints → but implemented as tether speed and acceleration constraints (?)

– implemented

Line 187 Results → Results and Discussion (?)

– implemented

Line 200 However, ... → please double-check language

– rewrote section

Line 202 Consider replacing the phrase "It is striking"

– replaced

Line 204 with in → within (?)

– implemented

Line 222 it's → its (double-check entire paper for this)

– implemented

Line 237 in higher drag losses and → in higher drag losses or

– implemented

Line 237 the cosine loss due elevation angle is not caused by gravity (remove "gravity-caused")

    – implemented

Line 265 what is meant by "maximum cycle-average loads"? → maximum load during a cycle?

    – removed sub-section

Line 292 "only cut-in wind speed" seems lost

    – removed

Line 335 lead result

    – removed 'lead'

Line 359 Determining ... determined

    – removed 'Determining'

Line 370 power only scales with the wing area ($F_{lift}\ b^2$) → note that the tether diameter and thus tether drag scale slower which is why power should scale faster than with $b^2$.

    – implemented

Line 427 sim → \sim

    – implemented

Line 457 a elliptical lift distribution → an elliptical lift distribution

– implemented

Line 471  do can not produce

    – fixed

Line 495  offshore AWES are not particularly beneficial relative to conventional wind, given the generally lower sheer offshore → Note that this is just another confirmation of the fact, that AWES advantage (at least for this concept), in particularly offshore, lies not in higher altitudes but reduced building material and associated benefits (transport etc.).

    – implemented

---

## Author Comment (AC6) · 16 Jan 2022

**Response to Short Comment 1 wes-2020-123**

Markus Sommerfeld

January 16, 2022

**1  Author response**

Dear Roderick Read,

Thank you very much for your positive feedback and advice to our manuscript, "Ground-generation airborne wind energy design space exploration", wes-2020-123. I am very sorry that I did not respond to your review sooner. My full-time work and family, together with a relocation to Japan, required my attention and time.

Taking into account your comment, I changed the title of the manuscript to "Scaling effects of rigid wing ground-generation airborne wind energy systems" to specify the kind of AWES the research focuses on. You are right that I did not describe other AWES concepts in this paper. I did so to limit the scope of this already lengthy work and I hope that the new title and introduction communicate this limited focus on ground-generation AWES.

Sincerely, Markus Sommerfeld

---

## Referee Report (RR1)

**Reviewer comments**

- **Manuscript title:** Scaling effects of fixed-wing ground-generation airborne wind energy systems
- **Authors:** Markus Sommerfeld, Martin Dörenkämper, Jochem De Schutter, and Curran Crawford
- **Reviewer comments:**

The proposed paper is a great contribution to the AWE technology, and it would be helpful for any team want to rescale their project. It can predict and demonstrate an estimation for the expected power and AEP based on an existing reliable quasi-steady state reference model. My recommendation for this paper is accepted with minor comments. It is suggested to have the following modifications:

- Add a section for the nomenclature in the beginning of the paper.
- Revise the paper one more time more carefully. Some typos are appearing at line 61 "from small to to utility", line 249 "the chosen initialization The optimization … ", line 388 "rated power. Beyound rated ….", etc …
- In page 3, line 70, it is recommended to move the paragraph "For a detailed description of the WRF model and clustering algorithm see Sommerfeld et al. (2020)." To the section 2.
- It is an optional for the author to add the mathematical model for the cited paper where you used as a reference model.

---

## Referee Report (RR2)

GENERAL COMMENTS

The paper investigates an interesting topic: how ground-gen airborne wind energy systems flight dynamics scale with size.

In my opinion, it still needs major revisions to be accepted. A lot of analyses are carried out to study too many different configurations (different mass scaling, sizes, aerodynamic coefficients, wind resources etc). I believe that all these configurations are overcomplicating an already complicated analysis and driving the reader away from key conclusions. Why not to have a reference design and modifying one parameter a time? It would be more educational.

Moreover, many optimal control problems are not converged. I think a journal publication cannot have non-converged simulations among the results. Those optimizations are not "outliers", but non meaningful solutions which should not be even there. The results should be physical and meaningful, otherwise all the discussions are worthless. I recommend spending way more time on making sure the optimization setup is working robustly before starting the analyses. I agree with most of the conclusions, but unfortunately, I do not trust the implementation of the methods enough to consider the results a solid proof of the conclusions.

SPECIFIC COMMENTS

- Line 6: comma at line 6 and line 8 should not be there. Also the two "-"at line 8.
- 13: missing a space
- 31: please check all the websites references. The dates are misleading.
- 121: Is there any reference for the reel-out and in speed? The results you show (e.g. Figure 6) seem extremely dependent on these values. Are these values scaling with size? I would expect that with a generator up scaling the drum diameter would also increase and thus these values, for the same drum angular velocity. Maybe you can add a comment here.
- 124: Can you elaborate a bit more on the tether diameter design? It is not clear to me what it the procedure. Is it an iterative procedure?
- 144: Eq. 4 is not necessary
- 146: The total drag coefficient does not depend only on these three parameters, as you mention in the sentence after. Please rephrase.
- 193: The citation goes before, when talking about the $CD^{tether}$ of the QSM.
- 195: "This leads to an underestimation of total tether drag at the aircraft" Not clear what "this" refers to.
- 197: I don't understand why, if you have a tether model, you don't apply all external forces (including weight and inertial) to the element's nodes. Can you explain? Are inertial forces on the tether included in the analysis?
- 202: This information has already been given in 3.1. Please do not repeat. It makes more sense here.
- 225: Can you give the two power harvesting factor values?

- Fig 4: Maybe these plots could go to an appendix as they are standards results and some discussions removed.
- 234: remove "The wind field is assumed to be constant for every optimization.": it is obvious and stated in the previous sentence
- 252: "because lift-to-tether drag ratio scales linearly with wingspan." wing area maybe.
- 294: There should be physical or modeling/optimization reasons for this. Please elaborate on it. It is important to give an interpretation to optimal trajectory, otherwise we cannot trust the results.
- 295: These considerations should be moved to future works.
- Fig 7: I don't understand why there are points with the same x coordinate.
- 334: The optimization is a deterministic process. If we change one parameter defining the optimization (e.g. wind speed), the optimal results should be a continuous function of this parameter. If we find a discontinuity, we need to understand why. Can the wind speed discretization be increased to have points closer to the outliers?
- 362: same comment as before on the discontinuities
- 368: maximum wind speed of the dataset (otherwise the maximum wind speed is related to extreme events)
- Tab 2: The overbar on P should not be there
- 396: maybe eq 9 and 10 can be removed to save space. they are well known.
- 408: wrong measurement units
- Fig 14: can you indicate in the legend at which wind speed the first plot is done?
- 560: should not be there

---

## Referee Report (RR3)

**Review:**

This is a review for the paper "Scaling effects of fixed-wing ground-generation airborne wind energy systems (2022)." As scaling up will inevitable, moving up to commercial AWES, a study on the scaling effects is a valuable contribution to the R&D community. The authors did some good work on showing the potential scaling effects. Also, the two reference wind profiles and their classifications are of great value to the AWE research community. However, I would like to give some feedback to further improve the quality and making it a great contribution to the AWE community.

General:

- The results might be far off from reality as the reference aircraft, the AP2, is a first demonstrator of a company. The mass scaling might therefore be significantly different from 2.7 – 3.3. The company had different goals in mind for this demonstrator then it would have for a commercial system. I would like to see this stated better in the paper.

- I miss in the conclusion a statement where the effect of the assumptions are stated, especially on the awebox side, so the reader knows again what should be taken into account when reading each of the separate conclusions.

Specific:

- P4, l83: Why is it sorted by the wind speed of 200m? The operational altitude you consider is higher.

- P6, l108: wast → vast

- P7, l110: can not → cannot

- P7, l112: How do you justify the feasibility? You use simplifications to set up the OCP right in awebox. Inelastic tether etc.

- P9, figure 3: It might be good to show which systems are measured, and which ones are estimated/hypothetical.

- P10, l194: Is the aircraft and ground station a tether mass point? If not, what is done with half the tether segment at the ground and at the kite, which is then not assigned to a tether mass. If not taken into account, this produces inaccuracies in the total weight of the tether.

- P10, l202: How are the reeling speed and acceleration constraints determined?

- P26, l468: As the purpose of this paper is to show trends when scaling, the explanation for the increase should be tested in this paper. Possibly by starting the optimisation from different points and see if they converge to the same. This way you might be able to spot other local minima or when it converges to the same, it might be a different cause.

- P27, figure 15: I miss the explanation of the oscillations that happen at higher wind speeds, sorry if I missed this, then clarifying it better could prevent that. Also, why D/L and not L/D like the conventional way. As the axis is in percent, it might confuse the reader it is actually a value with no unit.

---

## Referee Report (RR4)

**Review (reviewer #3):**

This is a second review for the paper "Scaling effects of fixed-wing ground-generation airborne wind energy systems (2022)." Thank you for the detailed response and explanations to my questions and of others, it definitely drastically improved the quality of the paper. However, I would like to give one very little recommendation as a response to your answer at "figure 15". I can follow your reasoning and in the end it is your choice for what you feel is better in communication but try to be consistent in Figure 14 and Figure 15. They communicate the same values but differ in the way of presenting, L/W and L/D & W/L and D/L. Also I think Figure 15 should not be a percentage, Newton/Newton is still a ratio with no unit. After implementing this, your paper seems ready for publication.

---

## Editor Decision (ED1)

**Scaling effects of rigid wing ground-generation airborne wind energy systems**

Markus Sommerfeld1, Martin Dörenkämper2, Jochem De Schutter3, and Curran Crawford1

[revised manuscript text omitted]

The structure of the paper is as follows. Section 2 summarizes the onshore and offshore wind resource as well as the clustering results. For a detailed description of the WRF model and clustering algorithm see (Sommerfeld, 2020). Section 3 briefly introduces the AWES model and optimization method as well as the implemented constraints and initialization. Section 4 compares the results for six AWES

70 sizes with three different mass scaling assumptions and two sets of non-linear aerodynamic coefficients. We present, inter dight alia, trajectories, power curves and annual energy production estimates for a representative onshore and offshore location. Finally, Section 5 concludes the article with an outlook and motivation for future work to continue to advance AWES towards commercial reality.

**2 Wind data**

This study considers representative 10 min onshore (northern Germany, lat:  $53^{\circ}10'47.00''$ N, lon:  $12^{\circ}11'20.98''$ E) and offshore wind data (FINO3 research platform, lat:  $55^{\circ}11,7'$ N, lon:  $7^{\circ}9,5'$ E) derived from 12 months of WRF simulations each. Both locations are highlighted by a black dot in figure 1.

Figure 1. Topography-map of northern Germany with the representative onshore (Pritzwalk) and offshore (FINO3) locations highlighted with a black dot.8

Both horizontal velocity components of the resulting mesoscale wind data set are <del>clustered</del> using a **k**-means clustering algorithm (Pedregosa et al., 2011). According to previous investigations (Sommerfeld, 2020), a small number of clusters with few representative profiles per cluster yield good power and AEP estimates at reasonable computational cost. Therefore, the wind velocity profiles were grouped into k = 10 clusters from which the 5th, 50th and 95th percentile (sorted by wind speed at 200 m) were implemented into the optimization algorithm as design points to cover the entire annual wind regime.

The resulting average wind velocity profiles for each of the ten clusters, also known as centroids, are shown in the top row of figure 2. For presentation purposes, only each centroid's wind speed magnitude, colored according to average wind speed up to 500 m, is shown. The complete set of clustered profiles profiles are shown in grey. The cluster average wind profile shapes show wind shears typically associated with unstable and stable conditions. They follow expected location-specific trends

4

with lower wind shear and higher wind speeds offshore (right) in comparison to onshore (left). The associated, color-coded annual centroid frequency is shown in the center. The bottom subfigures summarize the wind speed probability distribution at a reference height of  $100 \le z \le 400$  m. We chose this reference height as a proxy for wind speed at operating altitude, because an a priori estimation is impossible, and onshore and offshore power curves are almost identical using this reference wind speed. For a detailed description of the WRF model and setup, the clustering process as well as the correlation between clusters and stability conditions see (Sommerfeld, 2020).

90

---

## Author Response (AR2)

**Response to editor wes-2020-123**

Markus Sommerfeld

January 2021

**Author response**

Dear editor,

Thank you very much for your very detailed and helpful comments to our manuscript, "Scaling effects of fixed-wing ground-generation airborne wind energy systems", wes-2020-123. Major revisions have been implemented into the manuscript and all the comments to our previous submission have been addressed and highlighted in the marked-up version. I hope that the paper will meet your standards and the review process can continue.

Sincerely, Markus Sommerfeld

---

## Author Response (AR3)

**Response to reviewers of wes-2020-123**

Markus Sommerfeld

July 15, 2022

**Author response**

Dear reviewers,

Thank you very much for your very detailed and helpful comments to our manuscript, "Scaling effects of fixed-wing ground-generation airborne wind energy systems", wes-2020-123.

Most comments have been implemented into the manuscript and highlighted in the marked-up version. We hope that the paper will meet your standards. The reviewers will receive individual responses to their comments.

Sincerely, Markus Sommerfeld

**Response to reviewer 1 of wes-2020-123**

Markus Sommerfeld

July 15, 2022

**Author response**

Dear reviewer 1,

Thank you very much for your very detailed and helpful comments to our manuscript, "Scaling effects of fixed-wing ground-generation airborne wind energy systems", wes-2020-123.

All, but 2 comments have been implemented in the latest version of the manuscript. Please see our response below.

Sincerely, Markus Sommerfeld

**1 Reviewer comments**

**1.1 Specific comments**

Line 6 Add a section for the nomenclature in the beginning of the paper.

– not implemented, not common for wes publications

Line 61 "from small to to utility",

– implemented

Line 249 "the chosen initialization The optimization ... ",

– implemented

Line 388 "rated power. Beyound rated ....", etc ...

– implemented

Line 70 , it is recommended to move the paragraph "For a detailed description of the WRF model and clustering algorithm see Sommerfeld et al. (2020)." To the section 2.

– implemented

- It is an optional for the author to add the mathematical model for the cited paper where you used as a reference model.

– not implemented to not increase the length of the paper even more.

**Response to reviewer 2 of wes-2020-123**

Markus Sommerfeld

July 15, 2022

**Author response**

Dear reviewer 2,

Thank you very much for your very detailed and helpful comments to our manuscript, "Scaling effects of fixed-wing ground-generation airborne wind energy systems", wes-2020-123.

We understand that the high amount of system configurations lead to a lot of content and complexity. But the contribution of this paper is to compare realistic performance of many different AWES configurations subject to realistic wind conditions. For that reason the initial title of the paper included a "design explaration" of rigid wing AWES.

There is a misunderstanding regarding non-converged solutions and "outliers". Non-converged solutions are not included in the results and plots, i.e. data point not displayed in figures. They are the result of infeasible boundary conditions e.g. too low wind speeds at which the AWES cannot operate. Outliers on the other hand, are feasible solutions that are within in the constraints of the optimization (e.g. tether tension, tether speed, operating conditions etc.) Most of these solutions are the result of the system de-powering above rated wind speed. In these cases the optimizer finds trajectories, which is within constrains, but differ from the expected trajectory in operating height, tether length etc. This becomes difficult for higher wind speeds because some constraints, e.g. tether tension, is active.

The manuscript went through several rounds of major revisions by multiple reviewers and the editor since its initial draft more than years ago. Most of the original code and result files were lost during a computer crash. As a result, much of the code had to be re-written and the computationally expensive and time consuming optimizations had to be re-run, after I moved to a new country. We would like to avoid running additional optimizations, because of the significant afford that is associated with additional optimizations and their post-processing.

Sincerely, Markus Sommerfeld

**1 Reviewer comments**

**1.1 Specific comments**

Line 6 comma at line 6 and line 8 should not be there. Also the two "-"at line 8.

- implemented

Line 13 missing a space

– implemented

Line 31 please check all the websites references. The dates are misleading.

– implemented

Line 121 Is there any reference for the reel-out and in speed? The results you show (e.g. Figure 6) seem extremely dependent on these values. Are these values scaling with size? I would expect that with a generator up scaling the drum diameter would also increase and thus these values, for the same drum angular velocity. Maybe you can add a comment here.

Line 124 Can you elaborate a bit more on the tether diameter design? It is not clear to me what it the procedure. Is it an iterative procedure?

Line 144 Eq. 4 is not necessary

– was put on request of previous reviewer

Line 146 The total drag coefficient does not depend only on these three parameters, as you mention in the sentence after. Please rephrase.

– implemented

Line 193 The citation goes before, when talking about the $CD^{tether}$ of the QSM.

– implemented and rephrased the tether model description

Line 195 "This leads to an underestimation of total tether drag at the aircraft" Not clear what "this" refers to.

– rephrased

Line 197 I don't understand why, if you have a tether model, you don't apply all external forces (including weight and inertial) to the element's nodes. Can you explain? Are inertial forces on the tether included in the analysis?

– rephrased the tether model description. All forces are applied to the nodes.

Line 202 This information has already been given in 3.1. Please do not repeat. It makes more sense here.

– implemented

Line 225 Can you give the two power harvesting factor values?

– implemented

Fig 4 Maybe these plots could go to an appendix as they are standards results and some discussions removed.

– not implemented, I'd rather keep them here

Line 234 remove "The wind field is assumed to be constant for every optimization.": it is obvious and stated in the previous sentence

– implemented

Line 252 "because lift-to-tether drag ratio scales linearly with wingspan." wing area maybe.

    – implemented

Line 294 There should be physical or modeling/optimization reasons for this. Please elaborate on it. It is important to give an interpretation to optimal trajectory, otherwise we cannot trust the results.

    – This is likely caused by the longer reel-in period required to return the aircraft to its inital position. Because the tether tension is consistently high during reel-out (Figure 7 (a)), the reel-out speed remains high as well, leading to a longer reel-out length (Figure 7 (f)).As a result, the last loop is "carried over" to the reel-in period, which would not be done for real deployed systems.

    – this only seems to happen for HL and high wind speeds. High wind speeds → high const force → no need to reduce reel-out speed to maintain high force → more reel-out length → longer reel-in distance to return to init pos → one loop gets carried over to reel-in period, long reel-in time

Line 295 These considerations should be moved to future works.

    – implemented

Fig 7 I don't understand why there are points with the same x coordinate.

    – Different wind speed profile shapes have similar reference wind speeds $U_{\mathrm{ref}}(100 \text{ m} \leq z \leq 400 \text{ m})$ which leads to multiple values on the same x coordinate.

Line 334 The optimization is a deterministic process. If we change one parameter defining the optimization (e.g. wind speed), the optimal results should be a continuous function of this parameter. If we find a discontinuity, we need to understand why. Can the wind speed discretization be increased to have points closer to the outliers?

    – The wind speed profiles implemented into the optimization are very different in shape and magnitude, in contrast to simple log profiles where it is possible to increase the fidelity. These discontinuities are due to the high variation in wind conditions. Completely different profile shapes, can have similar average wind speed ($U_{ref}$). By increasing the amount of optimized wind profiles we would find that the results form are within a bandwidth from an average curve.

    – Added a line in the beginning of the "Results and discussion" section: The dotted lines that connect the data points are only their to better visualize the data and do not indicate a smooth continuity between data points.

    – if this does not satisfy your comment we could remove the dashed and dotted lines in between data points, but we would prefer to keep them.

Line 362 same comment as before on the discontinuities

    – Same as above

Line 368  maximum wind speed of the dataset (otherwise the maximum wind speed is related to extreme events)

– implemented

Tab 2  The overbar on P should not be there

– added overbar over other P for power since it is the cycle-average power

Line 396  maybe eq 9 and 10 can be removed to save space. they are well known.

– yes, they are well known, but I would like to keep them in to avoid ambiguity

Line 408  wrong measurement units

– implemented

Fig 14  can you indicate in the legend at which wind speed the first plot is done?

– clarified. These figures show the averaged weight and drag over the entire wind speed range to give a general trend.

– More details on how these results evolve over reference speed can be found in Figure 15

Line 560  should not be there

– implemented

**Response to reviewer 3 of wes-2020-123**

Markus Sommerfeld

July 15, 2022

**Author response**

Dear reviewer 3,

Thank you very much for your very detailed and helpful comments to our manuscript, "Scaling effects of fixed-wing ground-generation airborne wind energy systems", wes-2020-123.
Your comments have been implemented in the latest version of the manuscript and wording has been refined. Please see our response below.

Sincerely, Markus Sommerfeld

**1 General comments**

- The results might be far off from reality as the reference aircraft, the AP2, is a first demonstrator of a company. The mass scaling might therefore be significantly different from 2.7 – 3.3. The company had different goals in mind for this demonstrator then it would have for a commercial system. I would like to see this stated better in the paper.

    – implemented

- I miss in the conclusion a statement where the effect of the assumptions are stated, especially on the awebox side, so the reader knows again what should be taken into account when reading each of the separate conclusions.

    – Model assumptions are stated in the beginning of the conclusion section.

**1.1 Specific comments**

Line l83 Why is it sorted by the wind speed of 200m? The operational altitude you consider is higher.

    – implemented

Line 108 wast → vast

    – implemented

Line 110 can not → cannot

– implemented

**Line 112** How do you justify the feasibility? You use simplifications to set up the OCP right in awebox. Inelastic tether etc.

– rephrased

**Figure 3** It might be good to show which systems are measured, and which ones are estimated/hypothetical.

– implemented

**Line 194** Is the aircraft and ground station a tether mass point? If not, what is done with half the tether segment at the ground and at the kite, which is then not assigned to a tether mass. If not taken into account, this produces inaccuracies in the total weight of the tether.

– rewrote tether section. Yes, half of the tether mass at each endpoint (node) is assigned to either the kite or the ground.

**Line 202** How are the reeling speed and acceleration constraints determined?

– implemented
– They are very similar to the constraints used in the references of the AP2 reference model (Ampyx, 2020; Licitra, 2018; Malz et al., 2019 - line 118). They were also confirmed during conversations with Ampyx and ground station manufacturers.

**Line 468** As the purpose of this paper is to show trends when scaling, the explanation for the increase should be tested in this paper. Possibly by starting the optimisation from different points and see if they converge to the same. This way you might be able to spot other local minima or when it converges to the same, it might be a different cause.

– Starting the optimization from different initial guesses is possible, but beyond the scope of this paper. Doing this for approx 30 profiles, 6 designs, 2 sets of aerodynamic coefficients and 2 locations is not feasible.

**Figure 15** I miss the explanation of the oscillations that happen at higher wind speeds, sorry if I missed this, then clarifying it better could prevent that. Also, why D/L and not L/D like the conventional way. As the axis is in percent, it might confuse the reader it is actually a value with no unit.

– Deviation from the expected trend lines are likely caused by the wind speed profile shapes and variations in optimized trajectory, especially towards higher wind speeds. To stay within the constraints, the optimizer determines trajectories which vary in shape, operating altitude and tether length from the typical trajectories found at lower wind speeds.
– Total drag also includes tether drag and is therefore not the typical L/D of an airplane. We chose to display D/L, to use a consistent denominator when plotting W/L and D/L. Furthermore, we believe that this presentation better communicates the weight and total drag limitation of AWES (for this design).

---

## Author Response (AR4)

**Response to reviewer 1 and 5 of wes-2020-123**

Markus Sommerfeld

August 10, 2022

**Author response**

Dear reviewers,

Thank you very much for your helpful comments to our manuscript, "Scaling effects of fixed-wing ground-generation airborne wind energy systems", wes-2020-123.
We implemented the demanded changes.

Sincerely, Markus Sommerfeld

**1 Reviewer comments**

**1.1 Specific comments - reviewer 1**

- Plots with more points at the same wind speed: Maybe it would be more appropriate to draw the line to the mean values at each wind speed, if more data have the same wind speed.

    - Re-plotted Figure 7.
    - Thank you for pointing this out again. Checking the code, I realized that there was an error which lead to the same data points from the AP2 aero coefficients to also be plotted for HL. I edited the tension trough code. The plot is now clearer and trends are easier to understand.

- Typos

    - fixed typos, removed and added spaces, changed figure labels, fixed unit style

- Space between numbers and their measurements units

    - added spaces between numbers and measurement units

**1.2 Specific comments - reviewer 5**

- Try to be consistent in Figure 14 and Figure 15. They communicate the same values but differ in the way of presenting, L/W and L/D & W/L and D/L.

    - Implemented. Now all plots show L/W and L/D

- Also I think Figure 15 should not be a percentage, Newton/Newton is still a ratio with no unit.

  - Implemented.